# Visual functions in children with craniopharyngioma at diagnosis: A systematic review

Myrthe A. Nuijts[1]☯*, Nienke Veldhuis[2]☯, Inge Stegeman[3], Hanneke M. van Santen[4,5], Giorgio L. Porro[3], Saskia M. Imhof[3‡], Antoinette Y. N. Schouten–van Meeteren[6‡]

**1** Department of Ophthalmology, University Medical Center Utrecht, Utrecht University, Utrecht, The Netherlands, **2** Faculty of Medicine, Utrecht University, Utrecht, The Netherlands, **3** Department of Ophthalmology, University Medical Center Utrecht, Utrecht, The Netherlands, **4** Department of Pediatric Endocrinology, Wilhelmina Children's Hospital, University Medical Center Utrecht, Utrecht, The Netherlands, **5** Princess Máxima Center for Pediatric Oncology, Utrecht, The Netherlands, **6** Department of Neuro-Oncology, Princess Máxima Center for Pediatric Oncology, Utrecht, The Netherlands

☯ These authors contributed equally to this work.
‡ SMI and AYNSM also contributed equally to this work.
* M.A.Nuijts@umcutrecht.nl

**Data Availability Statement:** All relevant data are within the paper and its Supporting Information files.

## Abstract

Childhood craniopharyngioma is a rare and slow growing brain tumour, often located in the sellar and suprasellar region. It commonly manifests with visual impairment, increased intracranial pressure and hypothalamic and/or pituitary deficiencies. Visual impairment in childhood adversely affects a child's daily functioning and quality of life. We systematically reviewed the literature to provide an extensive overview of the visual function in children with craniopharyngioma at diagnosis in order to estimate the diversity, magnitude and relevance of the problem of visual impairment. Of the 543 potentially relevant articles, 84 studies met our inclusion criteria. Visual impairment at diagnosis was reported in 1041 of 2071 children (50.3%), decreased visual acuity was reported in 546 of 1321 children (41.3%) and visual field defects were reported in 426 of 1111 children (38.3%). Other ophthalmological findings described were fundoscopic (32.5%) and orthoptic abnormalities (12.5%). Variations in ophthalmological testing methods and ophthalmological definitions precluded a meta-analysis. The results of this review confirm the importance of ophthalmological examination in children with craniopharyngioma at diagnosis in order to detect visual impairment and provide adequate support. Future studies should focus on long-term visual follow-up of childhood craniopharyngioma in response to different treatment strategies to provide insight in risks and ways to prevent further loss of vision.

## Introduction

Childhood craniopharyngioma (CP) is a rare and slow growing epithelial brain tumour (World Health Organization grade I) [1]. It is thought to arise from embryonic remnants of Rathke's pouch, located along the craniopharyngeal duct. CP is commonly located in the sellar and/or suprasellar region of the brain [2, 3]. The incidence of CP is 0.5–2.0 new patients per

**Funding:** The PhD track of drs. M.A. Nuijts on visual impairment in children with a brain tumour is funded by a research grant from the Stichting Kinderen Kankervrij (KiKa) (grant number 304). This charitable foundation has no role in study design or conduct of this study, data collection, data analysis and interpretation or manuscript preparation. There is no contribution of commercial organizations.

**Competing interests:** The authors have declared that no competing interests exist.

**Abbreviations:** BCVA, best corrected visual acuity; CS, case series; CF, counting fingers; CP, craniopharyngioma; GKSR, gamma knife stereotactic radiosurgery; HM, hand motion; ICP, intracranial pressure; MC, multicenter; NA, not applicable; NPL, no perception of light; NR, not reported; OU, both eyes; PL, perception of light; PS, prospective; P-32, phosphorus-32; RS, retrospective; SC, single-center; UCVA, uncorrected visual acuity; UK, United Kingdom; USA, United States of America; VA, visual acuity; VF, visual field; VI, visual impairment.

million persons per year, with a bimodal distribution in children (5–14 years) and adults (50–74 years) [2–5].

Despite the benign histological grade I classification, CP often recurs and may cause severe morbidity due to its close anatomic relation with important visual and endocrinological structures. Affected children commonly present with visual impairment, increased intracranial pressure (ICP) and hypothalamic and/or pituitary deficiencies [2, 3, 6]. Impaired visual function is a primary manifestation in 62–84% of all children diagnosed with a CP [3]. Nevertheless, it often takes years after the onset of symptoms before children actually get diagnosed [4, 6].

Craniopharyngioma mainly causes visual impairment by direct infiltration and/or compression of the visual pathway. Damage of the visual pathway commonly manifests as decreased visual acuity (VA), visual field (VF) defects, typically bitemporal hemianopia, and/or abnormal pupillary reponses [5–11]. Increased intracranial pressure as a result of obstructive mass effect of the tumour can lead to papilledema with subsequent optic atrophy and permanent vision loss as potential complications [8, 12, 13]. In addition, therapeutic interventions for CP such as tumour resection or post-surgical radiation therapy can lead to further visual loss. In particular, gross total tumour resection has a high risk of visual loss as a result of direct damage to visual structures or disruption of its vascularisation [2–4, 9, 14–16].

As described above, CP and its therapy commonly causes severe and permanent visual impairment, as well as hypothalamic-pituitary dysfunction. These tumour sequelae have a major impact on a child's health and quality of life [12, 13, 17–19]. Therefore, early detection of visual impairment together with adequate treatment and support is of major relevance as it may reduce irreversible visual sequelae and improve long-term visual outcome and quality of life [12, 13, 20].

Early detection of visual abnormalities requires timely referral to an ophthalmologist for ophthalmological examination. Previous studies have already demonstrated the importance of ophthalmological examination in children with a brain tumour at time of diagnosis and during follow-up [12, 13, 17, 18, 20].

Several nonsystematic reviews have summarized ophthalmological findings in children and adults with CP. However, an extensive overview in subtopics like VA, VF, fundoscopy and orthoptic examination has never been published. With this systematic review we aim to provide a broad overview of the visual function in children with CP at diagnosis in order to estimate the diversity, magnitude and relevance of the problem of visual impairment in children with CP.

## Methods

### Protocol and registration

A review protocol was developed based on the Preferred Reporting Items of Systematic Reviews and Meta-Analyses (PRISMA) statement [21]. The systematic review was prospectively registered in the international prospective register of systematic reviews (PROSPERO) on April 23, 2020 (ID: 150419). In accordance with Dutch guidelines, no institutional ethical review board approval was required.

### Information sources and search strategy

We conducted a systematic search in the Cochrane Library, Embase and PubMed in order to identify all eligible studies. The electronic databases were last searched on October 2, 2019 for a combination of the following key search terms and/or their synonyms: 'craniopharyngioma', 'vision', 'visual acuity', 'visual fields', 'optic chiasm', 'optical coherence tomography', fundoscopic abnormalities (e.g. 'papilledema') and orthoptic abnormalities (e.g. 'diplopia').

The full search strategies are presented in S1 Appendix. We did not apply date, language or publication status restrictions. We limited search terms to presence in title or abstract to reduce the number of irrelevant articles. Reference lists of the included studies were reviewed for possible relevant articles. We did not search any trial registries for unpublished trials and no study authors were contacted to identify additional studies. All records identified were managed using Rayyan QCRI [22].

## Study selection

Two authors (M.N. and N.V.) independently screened titles and abstracts of studies identified from the electronic searches for potential inclusion. Full-text articles were obtained from potentially relevant abstracts and were assessed for eligibility by the two authors. Discrepancies were resolved by discussion. Both review authors were unmasked to article authors, journal, institution and trial results during the assessment.

## Eligibility criteria

We included all study types except case reports in which < 2 patients were included. Studies were included if patients were diagnosed with a CP and if data from children could be specifically extracted. Studies including patients who had received treatment before study participation and/or had recurrent CP were excluded.

## Outcome measures

The primary outcome measures of this systematic review were the presence of visual impairment, VA and VF at diagnosis. Secondary outcome measures of our study were results of fundoscopy and orthoptic examination at diagnosis.

## Risk of bias in individual studies

Risk of bias of the included studies was assessed by two reviewers (M.N. and N.V) independently of each other, using the Newcastle-Ottawa Scale (NOS) [23]. Any discrepancies between the reviewers were resolved by discussion. The NOS uses a star rating system for risk of bias assessment of three main parameters: selection and definition of study groups; comparability of study groups; and outcome assessment. The star ranking method in our review was based on predefined criteria, in which a total of 7 stars could be awarded. A detailed description of the risk of bias assessment is given in Table 1.

## Data analysis and synthesis

All data from the included studies were extracted in duplicate by two authors (M.N. and N.V.) independently. A standard data extraction form was used, including study characteristics (e.g. study size, study design, age, gender, tumour location) and outcome measures (e.g. VA, VF, fundoscopy and orthoptic examination) with associated outcome definitions if available. We quantified the extracted data per item and presented numbers for each item in two tables. Variations between studies and its outcome measures precluded a meta-analysis.

# Results

## Study selection

We identified 3653 records through the literature search in PubMed, Embase and the Cochrane Library. After removal of duplicates, 2372 records were screened by title and

**Table 1. Detailed description of risk of bias assessment using Newcastle-Ottawa Scale (NOS).**

| Domains | | Predefined criteria | Maximum number of stars per domain |
|---|---|---|---|
| Selection | Representativeness of exposed cohort (children with CP and visual impairment) | * Cohort truly representative of the average child with a primary CP aged 0–18 or 0–21 years in the community together with a description of key characteristics (age, gender, tumour type etc.)<br>• Selected group of children with CP (e.g. only giant CP)<br>• No description of key characteristics | **** |
| | Selection of non-exposed cohort (children with CP without visual impairment) | * Cohort drawn from the same community as the exposed cohort<br>• Cohort drawn from a different source<br>• No description of the derivation of the non-exposed cohort | |
| | Ascertainment of exposure (CP) | * Medical records / histological confirmation<br>• No description | |
| | Demonstration that outcome of interest (visual impairment) was not necessarily present at start of study | * Outcome of interest was not an inclusion criterion for study<br>• Outcome of interest was an inclusion criterion for study | |
| Comparability | Comparability of cohorts on the basis of the design of analysis | NA or for studies with > 1 cohort:<br>* Only children (aged 0–21 years) included in both cohorts<br>* Tumour locations were reported in both cohorts | NA or ** |
| Outcome | Assessment of outcome | * VA and VF were reported<br>• Only global information about visual function at diagnosis | * |
| | Was follow-up long enough for outcomes to occur | NA | |
| | Adequacy of follow-up of cohorts | NA | |

CP: Craniopharyngioma. NA: Not applicable. Items do not apply to the research question and design of this review.

abstract. In total, 494 full-text articles should be assessed for eligibility. However, full-text articles of 117 potential relevant abstracts were not available in the electronic databases. In attempt to retrieve these full-text articles, we searched the Utrecht University Library, Sci-Hub and contacted the corresponding author by mail and/or ResearchGate. Of these 117 potential relevant abstracts, 62 abstracts were published between 1956 and 2000 and 55 abstracts were published between 2001 and 2018. Finally, 377 full-text articles were assessed for eligibility. Together with 15 studies identified through reference screening [24–38], this resulted in 92 studies eligible for inclusion. However, we found that 18 studies reporting about patients diagnosed and treated for CP in the following hospitals (Hospital for Sick Children, Toronto; Boston Children's Hospital; General Navy Hospital, Beijing; Hospital National de Pediatria 'Prof. Juan P. Garrahan'; Institute of Neurological Sciences, Glasgow; Hôspital Necker-Enfants Malades; Great Ormond Street Children's Hospital; Johns Hopkins Hospital). Of these 18 studies, sixteen studies had overlapping periods of patient inclusion [10, 14, 29, 31, 39–50]. To avoid double inclusion of single patients, we decided to exclude the studies with the shortest period of patient inclusion and/or the least availability of visual data. This resulted in the exclusion of the following 8 studies: Abrams (1997) [44]; Banna (1973) [50]; Cohen (2013) [36]; Hetelekidis (1993) [45]; Sainte-Rose (2005) [40]; Thompson (2005) [41]; Yu (2015) [46] and Zuccaro (2005) [49].

Finally, 84 studies met the eligibility criteria and were included in our review. A detailed overview of the selection process for included studies and reasons for exclusion after full-text screening is shown in Fig 1.

## Study characteristics

Characteristics of the 84 included studies are shown in detail in Tables 2 and 3. Altogether, the studies included a total of 3531 children with CP with sample sizes ranging from 2 to 411

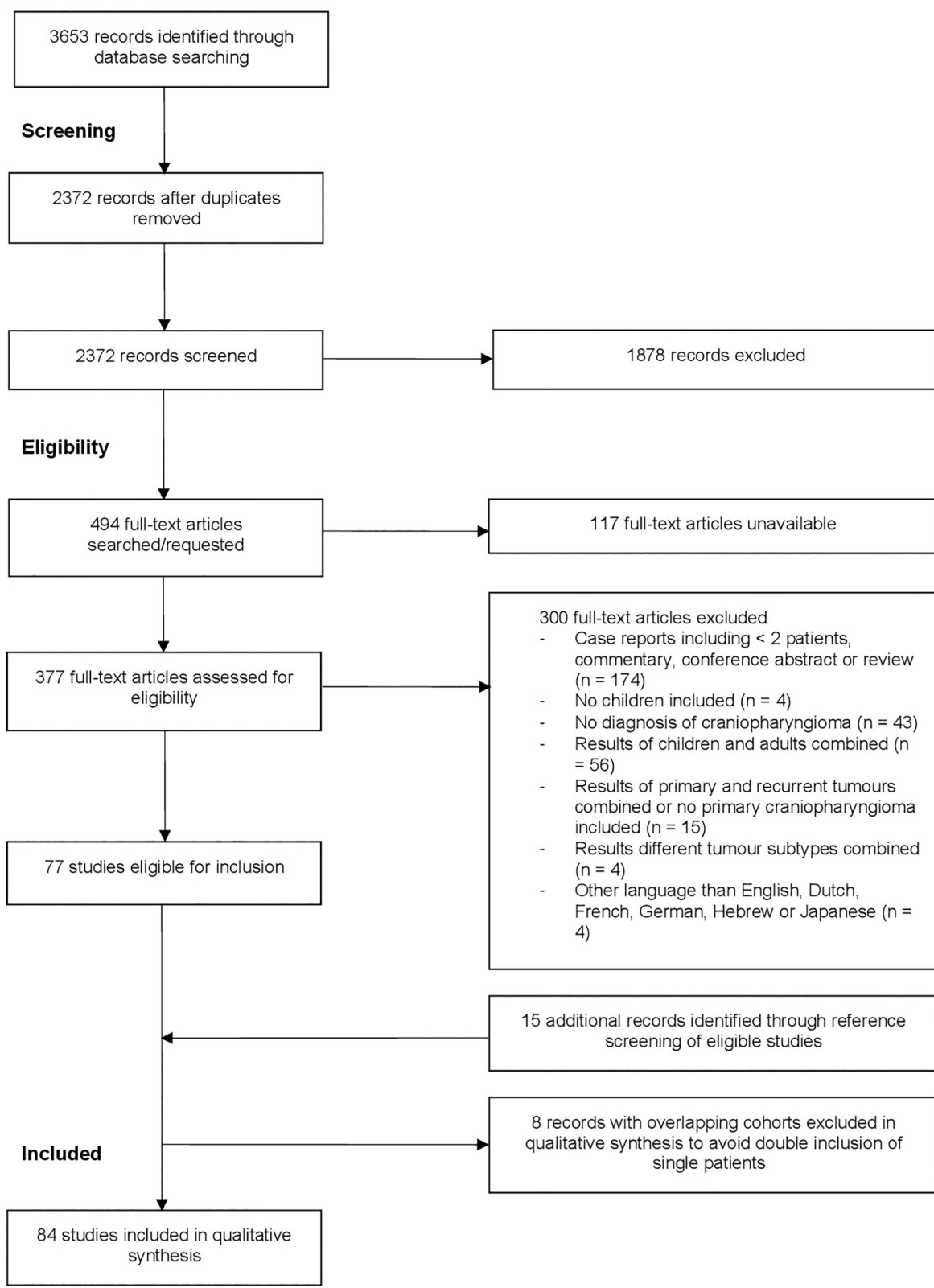

**Identification**

3653 records identified through database searching

**Screening**

2372 records after duplicates removed

2372 records screened → 1878 records excluded

**Eligibility**

494 full-text articles searched/requested → 117 full-text articles unavailable

377 full-text articles assessed for eligibility →

300 full-text articles excluded
- Case reports including < 2 patients, commentary, conference abstract or review (n = 174)
- No children included (n = 4)
- No diagnosis of craniopharyngioma (n = 43)
- Results of children and adults combined (n = 56)
- Results of primary and recurrent tumours combined or no primary craniopharyngioma included (n = 15)
- Results different tumour subtypes combined (n = 4)
- Other language than English, Dutch, French, German, Hebrew or Japanese (n = 4)

77 studies eligible for inclusion

15 additional records identified through reference screening of eligible studies

8 records with overlapping cohorts excluded in qualitative synthesis to avoid double inclusion of single patients

**Included**

84 studies included in qualitative synthesis

**Fig 1. PRISMA flow chart for identification and selection of studies.**

**Table 2. General characteristics of the included studies.**

| | Study | Study design, setting and country | Number of children, subtype if available | Mean age (years) at diagnosis, range/SD (years) | Gender (M/F) | Tumour location |
|---|---|---|---|---|---|---|
| 1 | Al-Mefty, 1985 [99] | RS, 1 cohort, SC, Saudi Arabia | 20 | (2–17), except one 23-year old man | 10/10 | NR |
| 2 | Albright, 2005 [51] | RS, 1 cohort, SC, USA | 44 | Micro neurosurgical tumor resection: 11, 12 (6–18)\*; P-32: 7, 7 (3–17)\*; GKSR: 13, 13 (5–18)\* | NR | NR |
| 3 | Ali, 2013 [54] | RS, 1 cohort, SC, USA | 7 | 9.6 (5–14) | 6/1 | Sellar and suprasellar 5 (71.4%), suprasellar 1 (14.3%), suprasellar and third ventricle 1 (14.3%) |
| 4 | Ammirati, 1988 [71] | RS, 1 cohort, SC, Germany | 3 | 13 (8–17) | 0/3 | Retrochiasmatic 4 (100%) |
| 5 | Anderson, 1989 [53] | RS, 1 cohort, SC, USA | 2 | 13.5 (12–15) | 2/0 | NR |
| 6 | Ansari, 2016 [62] | RS, 1 cohort, SC, USA | 9 | 6.7 (3–15) | 6/3 | NR |
| 7 | Artero, 1984 [107] | RS, 1 cohort, SC, Spain | 24 | 24 patients < 20 years | NR | NR |
| 8 | Ashkenazi, 1990 [82] | RS, 1 cohort, SC, Israel | 12 | NR (only for children & adults together) | NR (for all ages: 11/9) | NR. For children & adults: sellar extension 19/20, third ventricular 14/20 |
| 9 | *Bartlett, 1971 [35]* | *RS, 1 cohort, SC, USA* | *30* | *< 15 years* | *NR (for all ages: 42/43)* | *NR* |
| 10 | Behari, 2003 [81] | RS, 1 cohort, SC, India | 2 | 13.5 (11–16) | 2/0 | Intraventricular 2 (100%) |
| 11 | Bialer, 2013 [84] | RS, 1 cohort, SC, Israel | 20 | 6.5, SD 3.88 | 10/10 | NR |
| 12 | Boekhoff, 2019 [70] | RS, 1 cohort (PS follow-up), MC, Germany | 218; Adamantinomatous | 9.5 (1.3–17.9)\*: Symptomatic CP 9.6 (1.3–17.9); Incidental CP 8.1 (3.7–15.2) | 104/114: symCP 101/113; incCP: 3/1 | SymCP: extrasellar 44 (20.6%), intra- and extrasellar 153 (71.5%), intrasellar 3 (1.4%), not applicable 14 (6.5%); IncCP: extrasellar 2 (50%), intra- and extrasellar 2 (50%) |
| 13 | Cai, 2019 [76] | RS, 1 cohort, SC, China | 5 | 9.4 (3–13) | 4/1 | NR |
| 14 | Caldarelli, 2005 [90] | RS, 1 cohort, SC, Italy | 52 | 9 (1.67–15.8) | 33/19 | Intrasellar 3 (5.8%), sellar/suprasellar with prominent prechiasmatic growth 24 (46.2%), retrochiasmatic/third ventricular 14 (26.9%), giant (with an extension into the middle and/or posterior cranial fossae) 11 (21.2%) |
| 15 | Capatina, 2018 [106] | RS, 1 cohort, SC, Romania | 35 | 12.6, SD 4.2 | 16/19 | NR |
| 16 | Chamlin, 1955 [52] | RS, 1 cohort, SC, USA | 18 | NR | NR | NR |
| 17 | Chen, 2003 [6] | RS, 1 cohort, SC, Australia | 17; 9 Squamous; 6 Adamantinomatous; 2 NR | 10 | NR (for all ages: 17/19) | Children & adults: suprasellar 35 (97.2%), sellar 1 (2.8%) |
| 18 | Cherninkova, 1990 [102] | RS, 1 cohort, SC, Bulgaria | 50 | 9.5 | NR | NR |

(*Continued*)

**Table 2.** (Continued)

| | Study | Study design, setting and country | Number of children, subtype if available | Mean age (years) at diagnosis, range/SD (years) | Gender (M/F) | Tumour location |
|---|---|---|---|---|---|---|
| 19 | d'Avella, 2019 [91] | RS, 1 cohort, SC, Italy | 8 | 10.8 (8–16) | NR (8/4 including patients who had been previously surgically treated) | Supradiaphragmatic preinfundibular 2 (25%), supradiaphragmatic preinfundibular suprasellar 1 (12.5), supradiaphragmatic retroinfundibular 1 (12.5) infradiaphragmatic intra-suprasellar 3 (37.5%), infradiaphragmatic intra para-suprasellar 1 (12.5%) |
| 20 | Drimtzias, 2014 [72] | PS and RS, 1 cohort, SC, UK | 20 | 7.3 (1.25–13.75) | 10/10 | Suprasellar 20 (100%) |
| 21 | Erşahin, 2005 [101] | RS, 1 cohort, MC, Turkey | 87 | 10.2 (1.67–18) | 51/36 | Suprasellar 57 (66%), extended to third ventricle 22 (25%), temporal fossa 1 (1%), anterior cranial fossa 1 (1%), retroclival 4 (5%), temporal and posterior cranial fossa 2 (2%) |
| 22 | Fisher, 1998 [43] | RS, 1 cohort, SC, USA | 30 Adamantinomatous | 8.5, SD 5.3. 8.2 (0.74–18.9)* | 14/16 | Suprasellar 14 (47%), suprasellar and sellar 16 (53%) |
| 23 | Fouda, 2019 [63] | RS, 1 cohort, SC, USA | 135 | 8.5* (1–21) | 70/65 | Suprasellar 135 (100%): sellar extension 53 (39%) and third ventricular extension 56 (41%) |
| 24 | Gautier, 2012 [73] | RS, 1 cohort, two-center, France | 65 | < 10 years: 5.5 (4–6)*; 10–18 years: 12.5 (11–15)* | NR | < 10 year: intrasellar 1 (3.1%), extrasellar 9 (28.1%), intra/extrasellar 24 (75%); 10–18 year: intrasellar 3 (10.3%), extrasellar 8 (27.6%), intra/extrasellar 18 (62.1%) |
| 25 | Gerganov, 2014 [67] | RS, CS, SC, Germany | 1 Adamantinomatous | 14 | NR | Suprasellar, retrosellar and intraventricular 1/1 |
| 26 | Goldenberg-Cohen, 2011 [83] | RS, CS, SC, Israel | 4 | 4.9 (2.5–7.1) | 2/2 | NR |
| 27 | *Gonc, 2004* [31] | *RS, 1 cohort, SC, Turkey* | *66* | *8.4 (0.33–16.2)* | *30/36* | *Supra- and intrasellar 38 (58.5%), suprasellar 26 (41%), intrasellar 1 (1.5%)* |
| 28 | Greenfield, 2015 [60] | RS, 1 cohort, SC, USA | 24 | 7 (2–17.8)* | 12/12 | NR |
| 29 | Haghighatkhah, 2010 [104] | RS, CS, NR, Iran | 5 | 8.2 (6–12) | 2/2 | Suprasellar and sellar 2 (40%), suprasellar 2 (40%), posterior cranial fossa 1 (20%) |
| 30 | Hakuba, 1985 [86] | RS, CS, SC, Japan | 3 | 7.7 (6–10) | 3/0 | Suprasellar 3 (100%) |
| 31 | *Hoff, 1972* [37] | *RS, 1 cohort, SC, USA* | *16* | *(0.18–13)* | *6/10* | *NR* |
| 32 | *Hoffman, 1977* [25] | *RS, 1 cohort, SC, Canada* | *48* | *N = 17: (2–6)* *N = 31: (7–16)* | *24/24* | *NR* |
| 33 | Hoffman, 1992 [89] | RS, 1 cohort, SC, Canada | 50 | At time of surgery: 9.39 (1.83–17.58) | 28/22 | Prechiasmatic 25 (50%), retrochiasmatic 23 (46%), sellar 2 (4%) |
| 34 | Hoffmann, 2015 [68] | RS, 1 cohort (PS follow-up), MC, Germany | 411 | NR | NR | Intrasellar 5 (1,2%), suprasellar 61 (14,8%), intra- and suprasellar 169 (41,1%). |
| 35 | *Honegger, 1999* [26] | *RS, 1 cohort, SC, Germany* | *30* | *NR* | *NR* | *NR* |
| 36 | Hoogenhout, 1984 [93] | RS, 1 cohort, SC, The Netherlands | 12 | (0–15) | 9/3 | Children & adults: intra- and extrasellar 7 (25%), extrasellar 22 (75%) |
| 37 | Im, 2003 [97] | RS, CS, SC (except one case who had been treated at another hospital), Korea | 6 | 10.7 (5–14) | 2/4 | Intrasellar (pre-chiasmatic) 6 (100%) |

(*Continued*)

**Table 2.** (Continued)

| | Study | Study design, setting and country | Number of children, subtype if available | Mean age (years) at diagnosis, range/SD (years) | Gender (M/F) | Tumour location |
|---|---|---|---|---|---|---|
| 38 | Jane Jr., 2010 [64] | RS, 1 cohort, SC, USA | 11 | 12.3 (7–16) | 6/5 | Sellar and/or suprasellar 10 (90.9%), third ventricle 1 (9.1%) |
| 39 | Jung, 2010 [98] | RS, 1 cohort, SC, Korea | 17 | 12 (4–18)* | 12/5 | Suprasellar 10 (58.8%), supra- and intrasellar 7 (41.2%) |
| 40 | Karavitaki, 2005 [48] | RS, 1 cohort, MC, UK | 42 | 10 (2.5–15)* available in n = 35 | 23/19 | Intrasellar 1 (2.9%), extrasellar 13 (38.2%), intra- and extrasellar 20 (58.8%) available in n = 34 |
| 41 | Kennedy, 1975 [39] | RS, 1 cohort, SC, UK | 14 | (0–13) | NR (for all ages: 22/23) | NR |
| 42 | Kiran, 2008 [79] | RS, case reports (2x), SC, India | 2 | 8 (6–10) | 1/1 | Suprasellar with extension into the third ventricle and subtemporal extension to the left side with posterior extension 1 (50%), suprasellar with extension into third ventricle and posterior fossa 1 (50%) |
| 43 | *Kramer, 1960 [38]* | *RS, CS, MC, UK* | *6* | *10 (6.5–14)* | *5/1* | *Suprasellar 4 (66.7%)* |
| 44 | Lee, 2008 [108] | RS, 1 cohort, SC, Taiwan | 66 | 8.02, SD 4.28; 7.0 (1.42–17.58)* | 40/26 | Intrasellar 27 (40.9%), third ventricular 39 (59.1%) |
| 45 | *Lena, 2005 [24]* | *RS, 1 cohort, SC, France* | *47* | *N = 7: < 3 years*<br>*N = 10: (3–5)*<br>*N = 18: (6–10)*<br>*N = 12: > 10 years* | *27/20* | *Intrasellar 14 (29.8%): anterosuperior extension 9, pure intrasellar 3 and intrasphenoidal extension 2; infundibulotuberous 25 (53.2%); pure intraventricular 1 (2.1%); global/giant 7 (14.9%)* |
| 46 | Leng, 2012 [58] | RS, 1 cohort, SC, USA | 3 | 10.3 (5–15) | 2/1 | Sellar, suprasellar, retrochiasmatic 2 (67%); sellar, suprasellar, subchiasmatic 1 (33%) |
| 47 | Merchant, 2002 [65] | RS, 1 cohort, SC, USA | 30 | 8.6 (1–15)* | 13/17 | NR |
| 48 | Mohd-Ilham, 2019 [7] | RS, 1 cohort, SC, Malaysia | 11 | 9.5 (3–16) | 7/4 | Suprasellar 11 (100%) |
| 49 | *Mottolese, 2001 [33]* | *RS, 1 cohort, SC, France* | *14* | *NR* | *NR* | *NR* |
| 50 | Nielsen, 2012 [103] | RS, 1 cohort, MC, Denmark | 39 | < 15 years | NR | Intrasellar 27 (75%), intrasellar only 1 (2.7%), parasellar 10 (28.6%), suprasellar 36 (97.3%) |
| 51 | *Ohmori, 2007 [28]* | *RS, 1 cohort, SC, USA* | *27* | *9.0 (2–17)* | *17/10* | *NR* |
| 52 | Ono, 1996 [87] | RS, 1 cohort, SC, Japan | 19<br>Adamantinomatous | 8.1 (2–17)* | 11/8 | Prechiasmatic 11 (45.8%), retrochiasmatic 7 (29.2%), sellar 1 (4.2%) |
| 53 | Pascual, 2018 [55] | RS, 1 cohort, SC, USA | 35<br>Adamantinomatous | 11.4 (5–18) | 18/17 | Suprasellar pseudointraventricular 7 (20.0%), infundibulo-tuberal 6 (15.8%), sellar/suprasellar-secondary intraventricular 15 (39.5%), sellar/suprasellar 6 (15.8%), all intracranial spaces (giant) 1 (2.63%)* |
| 54 | Patel, 2017 [56] | RS, 1 cohort, SC, USA | 10 | 11.5 (5.9–15.0) | 6/4 | Sellar- and suprasellar 4 (40%), sellar, suprasellar and intraventricular 1 (10%), sellar, suprasellar and subchiasmatic 1 (10%), suprasellar 3 (30%), extracranial, infrasellar, nasal cavity and sphenoid sinus 1 (10%) |
| 55 | Puget, 2007 [14] | RS and PS, 2 cohorts, SC, France | 88 | RS cohort: 7.4 (1–16)*;<br>PS cohort: 8 (2.8–14)* | RS cohort: 42/24;<br>PS cohort: 13/9 | Prechiasmatic 20 (30%), retrochiasmatic 60 (91%), intraventricular 45 (68%), intrasellar 46 (70%) |

*(Continued)*

**Table 2.** (Continued)

| | Study | Study design, setting and country | Number of children, subtype if available | Mean age (years) at diagnosis, range/SD (years) | Gender (M/F) | Tumour location |
|---|---|---|---|---|---|---|
| 56 | Qi, 2012 [77] | RS, 1 cohort (2 subgroups), SC, China | 81 | Group A: 8.26 ± 4.03; | Group A: 23/11 | Group A: intra- and infrasellar 34 (42.0%); |
| | | | | Group B: 9.15 ± 3.83 | Group B: 26/21 | Group B: third ventricle: 47 (58%) |
| 57 | Quon, 2019 [57] | RS, 1 cohort, SC, USA | 16 | 11.8 (5.9–16)* | 11/5 | Suprasellar extension 15 (94%), NR 1 (6%) |
| 58 | Rath, 2013 [96] | RS, 1 cohort, SC, Australia | 10<br>Adamantinomatous | 9.4 (2.4–17.6) | NR | NR |
| 59 | Richmond, 1980 [61] | RS, 1 cohort, SC, USA | 21 | 0–4 years: 4/21, 5–8 years: 4/21, 9–12 years: 8/21, 13–16 years: 1/21, 17–20 years: 4/21 | 11/10 | NR |
| 60 | Salunke, 2016 [80] | RS, case reports, SC, India | 2 | 12.3 (8–18) | 2/1 | Suprasellar 2 (66%); suprasellar with erosion of sphenoid bone 1 (33%) |
| 61 | *Sankhla, 2015 [27]* | *RS, cohort, SC, India* | *6* | *13 (10–15)* | *4/2* | *NR* |
| 62 | Shammari, 2012 [100] | RS, 1 cohort, SC, Saudi Arabia | 2 | NR (only for children & adults together) | 1/1 | Suprasellar 2 (100%) |
| 63 | Shi, 2017 [78] | RS, 1 cohort, SC, China | 348<br>Adamantinomatous | 9.17 | 218/130 | NR |
| 64 | Sogg, 1977 [109] | RS, CS, center NR, country NR | 2 | 7.5 (6–9) | 0/2 | Third ventricle, tuber cinereum and the pituitary stalk 1 (50%), NR 1 |
| 65 | *Stahnke, 1984 [29]* | *RS, cohort, SC, Germany* | *28* | *8.4, SD 3.3 (2.5–14.5)* | *17/11* | *Suprasellar 13 (46.4%), intra- and suprasellar 11 (39.3%), intrasellar 4 (14.3%)* |
| 66 | Suharwardy, 1997 [9] | RS, cohort, SC, UK | 5 | NR | NR | NR |
| 67 | Synowitz, 1977 [69] | RS, CS, SC, Germany | 3 | 8.3 (6–12) | NR | Sellar 1 (33%), suprasellar 1 (33%), NR 1 (33%) |
| 68 | Tamasauskas, 2014 [105] | RS, CS, SC, Lithuania | 9<br>Adamantinomatous 8, Papillary 1 | 8.0 (0.83–17) | 6/3 | Suprasellar 3 (33%), suprasellar and intrasellar 3 (33%), suprasellar, intrasellar and parasellar 1 (11%), suprasellar, parasellar and retrosellar 1 (11%), suprasellar and parasellar 1 (11%) |
| 69 | Tan, 2017 [42] | RS, 3 cohorts, MC, UK | 185 | Group A: 6.60 (1.00–16.40), 7.70 (NA), Group B: 8.60 (7.23–9.99), 9.01 (3.83–16.00), 10.6 (3.50–16.20), 10.80 (1.50–15.50) | 85/100 | NR |
| 70 | Taphoorn, 2002 [94] | RS, CS, SC, The Netherlands | 3 | 10.3 (6–13) | 2/1 | Foramen intraventriculare 1 (33%), suprasellar with chiasm compression & enlargement ventricles 1 (33%), sellar 1 (33%) |
| 71 | Taylor, 2012 [74] | RS, 1 cohort, SC, France | 56 | 7.5 (6.6–8.5);<br>7.0 (4.9–9.9)* | 36/20 | NR |
| 72 | *Thomsett, 1980 [32]* | *RS, 1 cohort, SC, USA* | *42* | *9.2 (1.8–17.2)* | *24/18* | *NR* |
| 73 | Tomita, 2005 [66] | RS, 1 cohort, SC, USA | 54 | 8.2 (0.92–16)* | 28/26 | NR |
| 74 | Villani, 1997 [92] | RS, 1 cohort (PS follow-up), SC, Italy | 27 | 11 (6–16) | NR | Intrasellar 2 (7.4%), intrasuprasellar 13 (48.2%), suprasellar extraventricular 7 (25.9%), intra- and extra-ventricular 4 (14.8%), intraventricular 1 (3.7%) |
| 75 | Vries de, 2003 [85] | RS, 1 cohort, SC, Israel | 36 | 9.2. 7.8 (0.3–22.2)* | 19/17 | NR |

*(Continued)*

**Table 2.** (Continued)

| | Study | Study design, setting and country | Number of children, subtype if available | Mean age (years) at diagnosis, range/SD (years) | Gender (M/F) | Tumour location |
|---|---|---|---|---|---|---|
| 76 | Wan, 2018 [10] | RS, CS, SC, Canada | 59 | 9.4 (0.7–18.0) | 27/32 | NR |
| 77 | Weiss, 1989 [59] | RS, 1 cohort, SC, USA | 31 | 9.9 (1–19)* | 12/19 | NR |
| 78 | Wijnen, 2017 [95] | RS, 1 cohort, SC, The Netherlands | 63 | 8 (5–12)* | 25/38 | Intrasellar 3 (5%), suprasellar 23 (38%), intra- and suprasellar 34 (57%) |
| 79 | *Winkfield, 2011 [34]* | *RS, 1 cohort, SC, USA* | *79* | *At initial treatment: 8.5 (0.8–24.7)** | *43/36* | *NR* |
| 80 | Yamada, 2018 [88] | RS, 1 cohort, SC, Japan | 45 | At time of surgery: 9.6 (0.8–17.9) | 37/28 | Subdiaphragmatic 26 (58%): 3 sellar, 23 suprasellar; supradiaphragmatic 19 (42%): 1 purely intraventricular |
| 81 | Yano, 2016 [19] | RS, 1 cohort, SC, Japan | 26 | 7.3 (4–14) | 10/16 | NR |
| 82 | Yu, 2015 [46] | RS, 1 cohort, SC, France | 15 | 6.9 (0.25–14) | 9/6 | Sellar region with extension to the posterior cranial fossa 15 (100%) |
| 83 | *Zhang, 2008 [30]* | *RS, 1 cohort, SC, China* | *202* | *9.3 ± 3.6 (1–15)* | *115/87* | *NR* |
| 84 | Zhou, 2009 [75] | RS, CS, SC, China | 5 | 12 (9–18) | 3/2 | Posterior fossa 5 (100%): sellar 5 (100%), extension to cerebellopontine angle 5 (100%), infraclivus 2 (40%) |

Studies in italics indicate studies retrieved by reference screening.

* Median age.

CS: case series. UK: United Kingdom. USA: United States of America. GKSR: gamma knife stereotactic radiosurgery. MC: multicenter. NR: not reported. PS: prospective. P-32: phosphorus-32. RS: retrospective. SC: single-center.

children, with a mean study sample of 42 children (median of 21 children). Studies were published between 1955 and 2019; 40 of the 84 studies were published in the past 10 years. Twenty-two studies were conducted in the United States of America (USA) [28, 32, 34, 35, 37, 49, 51–66]; 7 studies in Germany [26, 29, 67–71]; 6 studies in the United Kingdom (UK) [9, 38, 39, 42, 48, 72] and France [14, 24, 33, 46, 73, 74]; 5 studies in China [30, 75–78]; 4 studies in India [27, 79–81], Israel [82–85] and Japan [19, 86–88]; 3 studies in Canada [10, 25, 89], Italy [90–92] and The Netherlands [93–95]; 2 studies in Australia [6, 96], Korea [97, 98], Saudi Arabia [99, 100], Turkey [32, 101]; 1 study in Bulgaria [102], Denmark [103], Iran [104], Lithuania [105], Malaysia [7], Romania [106], Spain [107] and Taiwan [108]; and for 1 study the country was not reported [109]. With regard to the included children with CP, 1236 were females and the mean age ranged from 0 [39, 93] to 23 years [99]. Data about gender or age were missing in respectively 19 [6, 9, 26, 33, 35, 39, 51, 52, 67–69, 73, 82, 91, 92, 96, 102, 103, 107] and 7 studies [9, 26, 33, 52, 68, 82, 100].

All studies had a retrospective study design, of which 5 studies also had a prospective follow up [14, 25, 70, 72, 92]. In most of the studies, a cohort of children was reviewed. These cohorts were generally obtained by screening medical records in single-center and in some studies multicenter hospitals. We also included twelve case series [10, 38, 69, 75, 83, 86, 92, 94, 97, 104, 105, 109] and two studies discussing more than one case report [79, 80].

There was a large variety between research aims of the included studies. In the majority of studies, visual function in children with CP at diagnosis was not a primary outcome measure, but details about visual function at diagnosis could be obtained from tables containing baseline characteristics. Precisely 10 studies reported primarily visual function and/or long term visual ouctomes in children with CP [6, 7, 10, 39, 52, 53, 72, 83, 102, 107].

**Table 3. Overview of visual function in children with craniopharyngioma at diagnosis.**

| | Study | Children with availability of vision data | Visual disturbance as symptom | Visual impairment | Decreased VA with description | Decreased VF with description | Orthoptic examination | Fundoscopy | Other vision related defects | Ophthalmological examination | Ophthalmological definitions |
|---|---|---|---|---|---|---|---|---|---|---|---|
| 1 | Al-Mefty, 1985 [99] | VA: 15/20, VF: 9/20 | NR | NR | 15/15: Severe decrease in both eyes 8/15, totally blind in both eyes 5/15, totally blind in one eye with decreased VA in the other eye 2/15 | Bitemporal defects 6/9, only central VF 1/9 | NR | Papilledema 7, optic atrophy 7, Foster-Kennedy syndrome 3 | NR | NR | NR |
| 2 | Albright, 2005 [51] | NR | NR | Intracavitary irradiation P-32: 4/44; Microneurosurgical tumor resection and GKSR: VI was one of the predominant neurological symptoms | NR | NR | NR | NR | NR | NR | NR |
| 3 | Ali, 2013 [54] | NR | 2/7 | 4/7 | Blurry vision 1/7 | NR | Diplopia 1/7 | NR | NR | Yes (not specified) | NR |
| 4 | Ammirati, 1988 [71] | 2/3 | 1/3 | 2/3 | 2/3: VA 20/100: 1/4 eyes, VA 20/200: 1/4 eyes, LP: 1/4 eyes | Bitemporal hemianopia 2/3 | NR | | NR | NR | NR |
| 5 | Anderson, 1989 [53] | 2/2 | 1/2 | 2/2 | 2/2: VA 20/25; CF at 1 foot 1/2, VA: 20/300; CF at 2 feet 1/2 | Homonymous hemianopia 1/2, temporal hemianopia 1/2 | NR | NR | NR | NR | NR |
| 6 | Ansari, 2016 [62] | NR | 5/9 | 5/9 | NR | NR | NR | NR | NR | NR | NR |
| 7 | Artero, 1984 [107] | VA: 21/24, VF: 18/24 | 5/24 | 22/24 | Decreased VA 17/21 | 16/18: Homonymous defects 5/18, temporal defects 14/18 (of whom 7/18 had bitemporal defects) | Diplopia 2/24 | Abnormal ocular fundus 19/24, optic atrophy or pallor 14/24 (unilateral 6/24), papilledema 7/24 | NR | VA and campimetric determinations, funduscopy and examination of ocular motility when the age and/or clinical condition allowed them to be performed. | NR |
| 8 | Ashkenazi, 1990 [82] | NR | 5/12 | NR | Decreased VA 5/12 | Reduction 3/12 | NR | NR | NR | NR | NR |
| 9 | Bartlett, 1971 [35] | NR | 23/30 | NR | NR | NR | NR | Papilledema 13/30 | NR | NR | NR |
| 10 | Behari, 2003 [81] | 2/2 | 2/2 | 2/2 | 2/2: 6/24: 1/4 eyes, 6/18: 2/4 eyes, 6/12: 1/4 eyes | NR | Bilateral sixth nerve palsy 1/2 | Bilateral papilledema 1/2 | NR | NR | NR |

(Continued)

**Table 3.** (Continued)

| | Study | Children with availability of vision data | Visual disturbance as symptom | Visual impairment | Decreased VA with description | Decreased VF with description | Orthoptic examination | Fundoscopy | Other vision related defects | Ophthalmological examination | Ophthalmological definitions |
|---|---|---|---|---|---|---|---|---|---|---|---|
| 11 | Bialer, 2013 [84] | 13/20 | 1/11 | 4/11 | ≤ 20/200 in at least one eye 7/11 | Bilateral temporal 4/15, unilateral temporal 3/15, right inferior homonymous quadrantopia 1/15 | RAPD 8/13, unilateral exotropia 6/13, sixth nerve palsy 2/13, monocular nystagmus 1/13, diplopia 3/13 | Papilledema 3/13, optic disc pallor 10/13: bilateral 7, unilateral 3 | NR | BCVA, Humphrey Field Analyzer | NR |
| 12 | Boekhoff, 2019 [70] | NR | Symptomatic CP 54/214 | Symptomatic CP 54/214, Incidental CP 1/4 | NR | Incidental CP: impaired VF right side 1/4 | NR | NR | NR | NR | NR |
| 13 | Cai, 2019 [76] | NR | NR | 3/5 | NR | NR | NR | NR | NR | VA and VF testing. | Visual outcome was graded as improved, stable, or deteriorated. |
| 14 | Caldarelli, 2005 [90] | NR | NR | 17/52 | Reduction of VA 13/52 | 9/52 | Sixth nerve deficit 5/52, third nerve deficit 1/52, nystagmus 2/52 | NR | NR | NR | NR |
| 15 | Capatina, 2018 [106] | NR | NR | NR | 22/35: Defect 11/35, decrease 15/35, uni- or bilateral blindness 7/35 | 18/35 | | NR | NR | Yes (not specified) | NR |
| 16 | Chamlin, 1955 [52] | NR | NR | NR | Loss of central VA 12/18 | Bitemporal hemianopia 18/18 | NR | Optic atrophy 14/18, papilledema 4/18 | Proptosis 1/18 | VF, optic discs, central VA (Snellen) and other ocular signs (extraocular muscle palsies, pupillary changes, involvement of NV, papilledema, proptosis, nystagmus) | As an indication of visual loss they took a very definite drop in VA (e.g. from a known 20/20 to 20/40 or less), or a reliable statement from the patient that his vision was definitely failing. |
| 17 | Chen, 2003 [6] | 16/17 | 15/17 | 13/17 | < 6/12: 6/17 (40%), ≥ 6/12: 10/17 (60%) | Bitemporal hemianopia 9/17, unilateral hemianiopia 1/17, homonymous hemianopia 1/17, normal VF 3/17, data NA 3/17 | Strabismus 3/17, RAPD 10/17 | Bilateral (optic) atrophy 10/17, bilateral papilledema 4/17 | NR | BCVA, Ishihara colour testing, RAPD, fundoscopy, cranial nerve examination, perimetry testing with Humphrey field analyser, Goldman perimetry or Bjerrum screen. | Normal if BCVA ≥ 6/12 |

(Continued)

**Table 3.** (Continued)

| | Study | Children with availability of vision data | Visual disturbance as symptom | Visual impairment | Decreased VA with description | Decreased VF with description | Orthoptic examination | Fundoscopy | Other vision related defects | Ophthalmological examination | Ophthalmological definitions |
|---|---|---|---|---|---|---|---|---|---|---|---|
| 18 | Cherninkova, [1990] [102] | NR | NR | NR | Reduced VA 32/46. Bilaterally reduced VA: under 0.1 6/50; over 0.1 17/50, amaurosis in one eye and reduced VA in the other 2/50, amaurosis in one eye and normal VA in the other 3/50, bilateral amaurosis 1/50, reduced VA in one eye and normal in the oter 3/50 | VF defects 21/31. Bitemporal hemianopia 5/31, amaurosis of one eye and temporal defect of the other eye 3/31, bilateral temporal narrowing of perimeters 5/31, homonymous hemianopia 2/31, bilateral concentric narrowing of the visual field 5/32, other defects 1/31 | Nystagmus 4/50, paresis of cranial nerve 6/50 | Optic nerve atrophy: unilateral 7/50, bilateral 19/50, congestive optic papilla 11/50 | NR | Ophthalmological examinations were performed by routine methods. In small children and patients in a serious condition a thorough study was not always possible. | NR |
| 19 | d'Avella, 2019 [91] | NR | 7/8 | NR | Left eye VA reduction 1/8 | Bitemporal hemianopia 3/8, bilateral superior quadrantopia 1/8, right temporal hemianopia 1/8 | NR | NR | Right amaurosis 1/8 | VA, computerized VF examination. | NR |
| 20 | Drimtzias, 2014 [72] | VA: 20/20 VF: 14/20 | 12/20 | 12/20 | 11/20: Mild-moderate 8/40 eyes, severe 13/40 eyes, normal 19/40 eyes | 10/14: bitemporal hemianopia 5/14 | NR | Optic atrophy 12/20, bilateral papilledema 6/20 | NR | BCVA (Logmar or Preferential Looking charts). VF with Goldmann perimetry | VA: normal (grade 8), mild-moderate visual loss (grade 5, 6, 7), severe visual loss (grade 1, 2, 3, 4). |
| 21 | Erşahin, 2005 [101] | NR | NR | NR | Blindness 13/87, visual disturbance and decreased VA 21/87 | 10/87 | Abducens paralysis 2/87, nystagmus 3/87, diplopia and squint 7/87 | Papilledema 5/87, optic atrophy 29/87 | NR | NR | NR |
| 22 | Fisher, 1998 [43] | NR | 19/30 | 19/30 | Loss of VA 17/30 | Loss of VF 14/30 | NR | NR | NR | NR | NR |
| 23 | Fouda, 2019 [63] | NR | 56/135 | 56/135 | Impaired VA 26/135 | Impaired VF 39/135 | NR | Papilledema 51/135 | NR | NR | NR |
| 24 | Gautier, 2012 [73] | 53/65 | 40/53 | 40/53 | Only reported together with VF 40/53 | Only reported together with VA 40/53 | NR | NR | NR | NR | Blindness: VA of 1/10 or less in both eyes. |
| 25 | Gerganov, 2014 [67] | NR | 1/1 | 1/1 | NR | NR | NR | NR | NR | Uniformly subjected preoperative ophthalmological assessment | NR |

(*Continued*)

**Table 3.** (Continued)

| | Study | Children with availability of vision data | Visual disturbance as symptom | Visual impairment | Decreased VA with description | Decreased VF with description | Orthoptic examination | Fundoscopy | Other vision related defects | Ophthalmological examination | Ophthalmological definitions |
|---|---|---|---|---|---|---|---|---|---|---|---|
| 26 | Goldenberg-Cohen, 2011 [83] | 3/4 | 2/4 | 3/4 | 3/4 | NR | Diplopia 1/4, esotropia 1/4, exotropia 1/4, mono-nystagmus 1/4 | Papilledema 1/4, optic atrophy 2/4, mild pallor 1/4 | NR | BCVA and a comprehensive neuro-ophthalmologic evaluation. | Severe visual loss was defined as counting fingers or less. |
| 27 | *Gonc, 2004* [31] | 64/66 | 23/66 | NR | 20/64: Unilateral 14/64, bilateral 6/64 | *Bitemporal hemianopia 22/64* | Diplopia 3/66 | Optic atrophy 27/64, papilledema 26/64 | NR | NR | NR |
| 28 | Greenfield, 2015 [60] | NR | NR | 21/24 | VA and VF deficits 16/24, legally blind (BCVA <20/200) 1/24 | VA and VF deficits 16/24 | Diplopia 8/24 | NR | NR | NR: only at follow-up. | Legally blind if corrected VA <20/200 in the better eye. |
| 29 | Haghighatkhah, 2010 [104] | NR | NR | 2/5 | Visual loss 1/5, left eye blindness 1/5 | NR | NR | NR | NR | NR | NR |
| 30 | Hakuba, 1985 [86] | NR | 1/3 | 1/3 | Failing vision 1/3 | Bitemporal hemianopia 1/3 | NR | Papilledema 2/3 | NR | NR | NR |
| 31 | *Hoff, 1972* [37] | 12/16 | NR | 7/12 | NR | NR | NR | *Papilledema 2/12, optic atrophy 7/12* | NR | *Satisfactory eye examination in 12/16 children.* | NR |
| 32 | *Hoffman, 1977* [25] | NR | 47/48 | 47/48 | *Significantly reduced unilaterally 17/48, significantly reduced bilaterally 11/48* | *Hemianopia 33/48: bitemporal 25/48, homonymous 4/48, unilateral temporal 4/48* | *Seesaw nystagmus 3/48* | *Papilledema 13/48* | NR | NR | *VA ≥ 20/40 bilaterally (mild visual loss); VA < 20/40 in one eye (moderate visual loss); VA < 20/40 bilaterally (severe visual loss).* |
| 33 | Hoffman, 1992 [89] | NR | 29/50 | 29/50 | Decreased in one or both eyes 21/50, blind in one eye 4/50 | 19/50: bitemporal hemianopia 8/50 | Diplopia 4/50, seesaw nystagmus 2/50 | NR | NR | No | NR |
| 34 | Hoffmann, 2015 [68] | 130/411 | 161/291 | NR | NR | NR | NR | NR | NR | NR | NR |
| 35 | *Honegger, 1999* [26] | NR | 10/30 | NR | NR | NR | NR | NR | NR | NR | NR |
| 36 | Hoogenhout, 1984 [93] | NR | 7/12 | NR | NR | NR | NR | NR | NR | Visual fields | NR |
| 37 | Im, 2003 [97] | NR | 5/6 | NR | NR | NR | NR | NR | NR | NR | NR |
| 38 | Jane Jr., 2010 [64] | NR | NR | NR | NR | 4/11 | NR | NR | NR | Formal visual field testing for patients with visual complaints. | NR |

(*Continued*)

**Table 3.** (Continued)

| | Study | Children with availability of vision data | Visual disturbance as symptom | Visual impairment | Decreased VA with description | Decreased VF with description | Orthoptic examination | Fundoscopy | Other vision related defects | Ophthalmological examination | Ophthalmological definitions |
|---|---|---|---|---|---|---|---|---|---|---|---|
| 39 | Jung, 2010 [98] | NR | NR | 4/17 | NR | NR | NR | NR | NR | Method used described by Fahlbusch and Schott to analyze ophthalmological findings (VA and VF). | NR |
| 40 | Karavitaki, 2005 [48] | 41/42 | NR | NR | Decreased 16/39 | 19/41 VF defects: bitemporal hemianopia 11/41 | NR | Papilledema 12/41, optic atrophy 2/41 | NR | NR | NR |
| 41 | Kennedy, 1975 [39] | VA 12/14, VF NR | NR | NR | Diminished vision 7/14 | 7/14 | Strabismus 5/14; rotatory nystagmus 1/14 | Optic atrophy 9/14, papilledema 6/14 | NR | VA, ocular movements, pupil reactions, ophthalmoscopy and VF testing using the Bjerrum screen. | NR |
| 42 | Kiran, 2008 [79] | 2/2 | 1/2 | 1/2 | 6/9 bilateral: 1/2 | NR | NR | Optic atrophy 1/2 | NR | VA, pupils and fundus examination | NR |
| 43 | Kramer, 1960 [38] | 6/6 | 3/6 | 3/6 | 2/6: 6/60: 1/6 | Bitemporal hemianopia 2/6, loss of right nasal field 1/6 | Right vertical and left rotatory nystagmus 1/6 | Papilledema 3/6, optic atrophy 3/6 | NR | Yes (not specified) | NR |
| 44 | Lee, 2008 [108] | NR | 21/66: Intrasellar 15/27; third ventricular 6/39 | NR | NR | NR | NR | NR | NR | NR | NR |
| 45 | Lena, 2005 [24] | NR | 32/47 | 32/47 | Blindness: bilateral 2/47, unilateral 3/47 | Pure VF defect 3/47, VF defect and decreased VA 23/47 | NR | Papilledema 13/47 | NR | Yes (not uniformly performed on all patients due to young age and emergency presentation). | NR |
| 46 | Leng, 2012 [58] | NR | 1/3 | 1/3 | NR | NR | NR | NR | NR | Neuro-ophthalmological evaluation and formal visual field testing when possible. | NR |
| 47 | Merchant, 2002 [65] | NR | 17/30 | NR | NR | NR | NR | NR | NR | NR | NR |

(Continued)

Table 3. (Continued)

| | Study | Children with availability of vision data | Visual disturbance as symptom | Visual impairment | Decreased VA with description | Decreased VF with description | Orthoptic examination | Fundoscopy | Other vision related defects | Ophthalmological examination | Ophthalmological definitions |
|---|---|---|---|---|---|---|---|---|---|---|---|
| 48 | Mohd-Ilham, 2019 [7] | 11/11 | 4/11 | 4/11 | BCVA ≥ 6/6-6/12: 13/22 eyes, BCVA 6/15-6/60: 3/22 eyes, BCVA < 6/60: 6/22 eyes | Temporal hemianopia 5/22 eyes: bilateral 2/22 eyes, unilateral 3/22 eyes; scotoma 3/22 eyes: central 2/22 eyes, inferior 1/22 eyes; quadrantanopia 2/22 eyes; constricted 1/22 eyes. VF is NA in 5 patients | Squint 2/11, diplopia 1/11, RAPD 7/11 | Optic atrophy 11/11, papilledema 2/11 | Color defect 4/11 | BCVA, VF (confrontational test or Humphrey visual field test), color vision, light brightness, RAPD, fundus examination and cranial nerves examination. | BCVA ≥ 6/12 (good) during presentation. Visual loss was defined as blurring of vision in both eyes. |
| 49 | *Mottolese, 2001* [33] | *NR* | *2/20* | *2/20* | *NR* | *NR* | *NR* | *NR* | *NR* | *NR* | *NR* |
| 50 | Nielsen, 2012 [103] | NR | NR | NR | Reduction 21/32, blindness 1/39 | Reduction 12/26 | NR | NR | Ophthalmoplegia 5/31 | VA and VF testing. | NR |
| 51 | *Ohmori, 2007* [28] | *NR* | *15/27 (55%)* | *NR* | *NR* | *NR* | *NR* | *NR* | *NR* | *For 21 patients only initial outcome data is available. 6 patients underwent extensive outcome analysis, including ophthalmological testing.* | *NR* |
| 52 | Ono, 1996 [87] | NR | NR | 15/19. Mean visual score at diagnosis 68.4. | NR | NR | NR | NR | NR | VA and VF testing. Visual scores were assigned in order to evaluate visual functions digitally assessing both VA and VF (0–100 points). | NR |
| 53 | Pascual, 2018 [55] | NR | 35/35 | 35/35 | NR | NR | Diplopia 14/35, sixth nerve palsy 1/35 | NR | NR | Yes (not specified) | NR |
| 54 | Patel, 2017 [56] | NR | 7/10 | 7/10 | NR | NR | NR | Papilledema 1/10 | NR | NR | NR |
| 55 | Puget, 2007 [14] | NR | RS cohort: 30/66 PS cohort: 14/22 | NR | Blindness 10/66 | NR | NR | NR | NR | NR | NR |
| 56 | Qi, 2012 [77] | NR | Group A: 34/34 Group B: 24/47 | 57/81 | Unilateral/ bilateral blindness or light perception: - Group A: 12/34 - Group B: 3/47. | NR | NR | NR | NR | NR | NR |

(Continued)

**Table 3.** (Continued)

| | Study | Children with availability of vision data | Visual disturbance as symptom | Visual impairment | Decreased VA with description | Decreased VF with description | Orthoptic examination | Fundoscopy | Other vision related defects | Ophthalmological examination | Ophthalmological definitions |
|---|---|---|---|---|---|---|---|---|---|---|---|
| 57 | Quon, 2019 [57] | NR | NR | 11/16 | Vision loss 4/16, blurry vision 2/16 | Bilateral hemianopia 1/16, bitemporal hemianopia 2/16; VF deficit 3/16 | NR | Papilledema 3/16, optic nerve compression 1/16 | Proptosis 1/16 | Complete work-up by an ophthalmologist when visual symptoms or signs were present. | NR |
| 58 | Rath, 2013 [96] | 10/10 | 4/10 | 7/10 | Mild VA deficit or field cut 4/10, unilateral blindness, homonymous hemianopia or bitemporal hemianopia 2/10, bilateral blindness or near functional blindness (unrelated) 1/10 | Mild VA deficit or field cut 4/10, unilateral blindness, homonymous hemianopia or bitemporal hemianopia 2/10, bilateral blindness or near functional 2 blindness (unrelated) 1/10 | Right exotropia 1/10 | Papilledema 1/10 | NR | NR | 1) Normal acuity and fields (3/10); 2) Mild acuity deficit or field cut (4/10); 3) Unilateral blindness, homonymous hemianopia or bitemporal hemianopia (2/10); 4) Bilateral blindness or near functional blindness. (1/10) |
| 59 | Richmond, 1980 [61] | NR | 9/21 | NR | Decreased VA: bilateral 2/21, unilateral 7/21 (3 blind). Unilateral blind 3/21. | 12/21 (3 of them had a combination of two findings). Temporal: unilateral 4/21; bilateral 3/21, homonymous 4/21, scotoma 1/21 | NR | Optic atrophy; unilateral 2/21; bilateral 5/21, papilledema 2/21 | NR | Yes (not further specified) | NR |
| 60 | Salunke, 2016 [80] | 2/2 | 2/2 | 2/2 | 2/2: 6/18: 2/4 eyes, 6/24: 1/4 eyes, PL plus 1/4 eyes | Bitemporal hemianopia 2/2 | NR | NR | NR | NR | NR |
| 61 | *Sankhla, 2015* [27] | NR | 3/6 | NR | *NR* | *NR* | *NR* | *NR* | *NR* | NR | *NR* |
| 62 | Shammari, 2012 [100] | NR | 2/2 | 1/2 | NR | NR | Rotatory nystagmus 1/2, horizontal pendular nystagmus 1/2 | Temporal disc pallor both eyes 2/2 | NR | Ophthalmic records were reviewed. | NR |
| 63 | Shi, 2017 [78] | NR | NR | 99/348 | NR | NR | NR | NR | NR | NR | NR |
| 64 | Sogg, 1977 [109] | 2/2 | 2/2 | 2/2 | 2/2: 20/100: 1/4 eyes, 20/50: 1/4 eyes, 20/400: 1/4 eyes, 20/200: 1/4 eyes | Bitemporal hemianopia 2/2 | NR | Papilledema 1/2, optic pallor 1/2 | NR | VA and VF testing (red test and large white test objects; Goldmann perimetry), fundoscopy. | NR |
| 65 | *Stahnke, 1984* [29] | NR | 12/28 | 24/28 | *Decreased VA 11/28* | *VF defect 16/28* | *NR* | *Optic atrophy 11/28, papilledema 4/28* | *NR* | *Ophthalmological examination.* | *NR* |

(*Continued*)

**Table 3.** (Continued)

| Study | Children with availability of vision data | Visual disturbance as symptom | Visual impairment | Decreased VA with description | Decreased VF with description | Orthoptic examination | Fundoscopy | Other vision related defects | Ophthalmological examination | Ophthalmological definitions |
|---|---|---|---|---|---|---|---|---|---|---|
| 66 Suharwardy, 1997 [9] | 5/5 | 1/5 | 5/5 | 6/24 2/10 eyes, NPL 1/10 eyes, 6/12 2/10 eyes, 6/9 1/10 eyes, 6/6 2/10 eyes, 1/60 1/10 eyes, HM 1/10 eyes | VF in the better eye: temporal defect 1/5, bitemporal hemianopia 1/5, asymmetric binasal loss with enlarged blind spots 1/5, right probable temporal loss 1/5, left supero-temporal loss 1/5 | RAPD 5/5 | Bilateral optic atrophy 1/5, papilledema 2/5, left disc pallor 1/5, bilateral disc pallor 1/5 | NR | A full ophthalmological examination including VA and VF testing (in most cases possible with a Snellen chart and Goldmann field respectively), optic discs, colour vision and pupil responses. | NR |
| 67 Synowitz, 1977 [69] | 3/3 | 3/3 | 3/3 | Fingerzahlen right/1 m and Handbewegung left/50 cm (1/3), LP (1/3), NR (1/3) | NR | Nystagmus 2/3 | Optic atrophy 2/3, papilledema 2/3 | NR | Yes (not further specified) | NR |
| 68 Tamasauskas, 2014 [105] | 7/9 | 4/9 | 4/9 | RE = LE = 5/10: 1, RE = 1 and LE = 1/1000: 1 | Bitemporal hemianopia 2/9, homonymous hemianopia 1/9 | NR | NR | NR | VA and VF testing before surgery and after surgery. | NR |
| 69 Tan, 2017 [42] | 136/185 | 90/136 | 90/136 | NR | NR | NR | NR | NR | VA and VF were assessed by experienced ophthalmologists. Children unable to cooperate were given a score based on visual evoked potentials. | NR |
| 70 Taphoorn, 2002 [94] | 3/3 | 1/3 | 3/3 | Decreased VA 1/3 | Bitemporal hemianopia 1/3, partial homonymous hemianopia 2/3 | Diplopia 1/3 | Papilledema 2/3 | NR | NR | NR |
| 71 Taylor, 2012 [74] | NR | 27/56 | NR | Reduced VA | NR | Strabismus, nystagmus | NR | NR | NR | NR |
| 72 Thomsett, 1980 [32] | NR | 15/42 | 15/42 | Decreased VA 14/33 | VF defect 13/32 | Cranial nerve palsy 12/42 | Optic atrophy 11/29, papilledema 9/42 | NR | NR | NR |
| 73 Tomita, 2005 [66] | NR | 23/54 | 23/54 | Decreased VA 13/54: monocular 10, binocular 11. | VF defect 2/54 | Diplopia 3/54, strabismus 2/54 | NR | NR | NR | NR |
| 74 Villani, 1997 [92] | NR | 16/27 | 16/27 | Decreased VA 16/27 | 11/27 | NR | NR | NR | NR | NR |
| 75 Vries de, 2003 [85] | NR | 8/36 | NR | NR | NR | NR | NR | NR | NR | NR |

(Continued)

**Table 3.** (Continued)

| | Study | Children with availability of vision data | Visual disturbance as symptom | Visual impairment | Decreased VA with description | Decreased VF with description | Orthoptic examination | Fundoscopy | Other vision related defects | Ophthalmological examination | Ophthalmological definitions |
|---|---|---|---|---|---|---|---|---|---|---|---|
| 76 | Wan, 2018 [10] | 59/59 | 18/59 | 25/59 | Visual impairment in at least 1 eye 25/59, binocular visual impairment 16/59, legally blind in both eyes 4/59 | NR | Diplopia or strabismus 7/59 | Optic nerve edema 25/59, optic nerve pallor 24/59 | NR | VA (preferential looking if vision too poor or pre-verbal) and VF testing (automated Humphrey, dynamic Goldmann or confrontation), fundoscopy. | Visual decline: defined as a move from a higher to lower category of visual function in 1 or 2 eyes. Visual outcomes were grouped normal, impaired and legally blind. |
| 77 | Weiss, 1989 [59] | NR | 24/31 | 24/31 | VA and/or VF 19/31 | VA and/or VF 19/31 | Sixth nerve deficit 4/31: unilateral 2, bilateral 2 | Asymptomatic papilledema or optic atrophy 5/31 | NR | NR | NR |
| 78 | Wijnen, 2017 [95] | VA: 46/63 VF: 39/63 | NR | NR | 33/46 | 23/39 | NR | NR | NR | VA was determined after correction for refraction disorders. Goldmann perimetry for VF testing. | NR |
| 79 | *Winkfield, 2011 [34]* | *NR* | 46/79 | *NR* | *NR* | *NR* | *NR* | *Papilledema 25/76* | *NR* | *NR* | *NR* |
| 80 | Yamada, 2018 [88] | 42/45 (3 patients could not be assessed due to their young age) | 12/45 | 28/42 | NR | NR | NR | NR | NR | Yes (except the youngest in whom testing was difficult): VA and VF testing before and 2 weeks after surgery. | NR |
| 81 | *Yano, 2016 [19]* | *NR* | 14/26 | *NR* | *NR* | *NR* | *NR* | NR | NR | | NR |
| 82 | Yu, 2015 [46] | 14/15 | NR | 13/15 | Decreased vision 13/14 | NR | Cranial nerve palsy 3/15 | NR | NR | | NR |
| 83 | *Zhang, 2008 [30]* | *NR* | *NR* | *113/202* | *NR* | *NR* | *NR* | *NR* | *NR* | *NR* | *NR* |
| 84 | Zhou, 2009 [75] | NR | 4/5 | 1/5 | NR | Hemianopia 1/5 | Diplopia 1/5, bidiplopia 1/5, dilated pupil on the left 2/5 | Bipapilledema 3/5 | NR | NR | NR |

Studies in italics indicate studies retrieved by reference screening.

BCVA = best corrected visual acuity. GKSR: gamma knife stereotactic radiosurgery. HM = hand motion. NA: not available. NPL = no perception of light. NR: not reported. OU = both eyes. PL = perception of light. P-32: phosphorus-32. VA: visual acuity. VF: visual field. VI: visual impairment.

## Risk of bias assessment

Table 4 shows the results of the risk of bias assessment by the Newcastle-Ottawa Scale (NOS) for the included studies. Overall, total scores ranged between 1 to 6 stars of a possible 7 stars. Eleven studies were awarded 1–3 stars, 72 studies 4–5 stars and one study 6 stars. These scores indicate that most of the included studies were of moderate quality. Quality was predominately limited by missing or incomplete information about VA testing and/or VF testing [14, 19, 26–28, 30, 33–35, 51, 54–56, 62, 67, 68, 70, 74, 77, 85, 93, 97, 100, 104, 108]. Other important reasons for weaker quality were studies who included only patients who are known to have an impaired visual function [83, 100], studies who included only patients who are known to have a CP at a specific location [46, 54, 71, 75, 76, 80, 81, 86, 108] and studies who included only patients with giant or extensive CP [46, 67, 79, 99, 104]. We were not able to score comparability for 82 studies, because no cohorts were compared in these studies. Puget (2007) [14] and Tan (2017) [42] were the only two studies we could rate for comparability, since they included two and three cohorts, respectively.

## Tumour location

Fourty-seven of the 84 studies described the CP location in a total of 1895 children (Table 2), although different anatomical terms, without strict definitions of terminology, were used. In 3 studies it was not clear if tumor location was concerned for child or adult CP: Ashkenazi (1990) reported 19 CP with sellar extension and 14 third ventricular CP [82], Chen (2003) reported 35 suprasellar CP and one sellar CP [6] and Hoogenhout (1984) reported 22 extrasellar CP and 7 intra- and extrasellar CP [93].

Craniopharyngioma was located (intra)sellar in 153 children (8.1%). In 34 children CP was located intra- and infrasellar (1.8%). Villani (1997) [92] reported 4 intra- and extraventricular CP (0.2%).

(Intra)sellar and suprasellar CP were reported in 244 children (15.9%) and sellar and/or suprasellar CP in 20 children (1.1%). Sellar, suprasellar and intraventricular CP were reported in 16 children (0.8%) [55, 56]. Extrasellar CP was reported in 76 children (4.0%) and intra- and extrasellar in 217 children (11.5%).

Suprasellar CP was reported in 477 children (25.2%), of which 27 CP were not purely suprasellar. Intraventricular CP was reported in 49 children (2.6%). Quon (2009) [57] reported 15 CP with suprasellar extension and one CP without tumour location. Seven patients had a suprasellar extraventricular CP [92]. In a study by Tamasauskas (2014) 2 of 9 children had respectively a suprasellar, intrasellar and parasellar CP and a suprasellar, parasellar and retrosellar CP [105]. Gerganov (2014) reported one suprasellar, retrosellar and intraventricular CP [67]. Taphoorn (2002) reported one suprasellar CP with enlargement of ventricles and chiasm compression [94]. Lastly, Kiran (2008) reported one suprasellar CP with extension to the third ventricle and subtemporal extension to the left side with posterior extension [79].

Craniopharyngioma was located third ventricular or extended to the third ventricle in 110 children (5.8%). Retrochiasmatic CP was reported in 94 children (5.0%), with a sellar and suprasellar component in 2 patients in a study by Leng (2012) [58]. Caldarelli (2005) reported 14 retrochiasmatic or third ventricular CP [90]. Twenty-four children (1.5%) had a CP located in or with extension to the posterior cranial fossa, namely 2 CP were located at the temporal and posterior cranial fossa [101], 1 suprasellar CP with extension to the posterior cranial fossa and third ventricle [79], 15 sellar CP with extension to the posterior cranial fossa [46] and 5 sellar CP with extension to the cerebellopontine angle and the posterior cranial fossa, as well as infraclivus extension in 2 of 5 patients [75].

Prechiasmatic CP was reported in 80 children (4.2%). Of these, 24 CP were sellar or suprasellar with prominent prechiasmatic growth [90]. D'Avella (2019) [91] reported 23

**Table 4. Risk of bias assessment for cohort studies using the New-Ottawa Scale (NOS).**

| Number | Study | Selection | | | | Comparability | Outcome | | | Total number of stars |
| --- | --- | --- | --- | --- | --- | --- | --- | --- | --- | --- |
| | | Represen-tativeness of exposed cohort | Selection of non-exposed cohort | Ascertainment of exposure | Demonstration that outcome of interest was not present at start of study | Comparability of cohorts on the basis of the design or analysis | Assessment of outcome | Was follow-up long enough for outcomes to occur | Adequacy of follow-up of cohorts | |
| 1 | Al-Mefty, 1985 [99] | | * | * | * | NA | * | NA | NA | 4 |
| 2 | Albright., 2005 [51] | * | * | * | * | NA | | NA | NA | 4 |
| 3 | Ali, 2013 [54] | | * | * | * | NA | | NA | NA | 3 |
| 4 | Ammirati, 1988 [71] | | * | * | * | NA | * | NA | NA | 4 |
| 5 | Anderson, 1989 [53] | * | * | * | * | NA | * | NA | NA | 5 |
| 6 | Ansari, 2016 [62] | * | * | * | * | NA | | NA | NA | 4 |
| 7 | Artero, 1984 [107] | * | * | * | * | NA | * | NA | NA | 5 |
| 8 | Ashkenazi, 1990 [82] | * | * | | * | NA | * | NA | NA | 3 |
| 9 | *Bartlett, 1971* [35] | * | * | * | * | *NA* | | *NA* | *NA* | *4* |
| 10 | Behari, 2003 [81] | | * | * | * | NA | * | NA | NA | 4 |
| 11 | Bialer, 2012 [84] | * | * | * | * | NA | * | NA | NA | 5 |
| 12 | Boekhoff, 2019 [70] | * | * | * | * | NA | | NA | NA | 4 |
| 13 | Cai, 2019 [76] | | * | * | * | NA | * | NA | NA | 4 |
| 14 | Caldarelli, 2005 [90] | * | * | * | * | NA | * | NA | NA | 5 |
| 15 | Capatina, 2018 [106] | * | * | * | * | NA | * | NA | NA | 5 |
| 16 | Chamlin, 1955 [52] | * | * | * | * | NA | * | NA | NA | 5 |
| 17 | Chen, 2003 [6] | * | * | * | * | NA | * | NA | NA | 5 |
| 18 | Cherninkova, 1990 [102] | * | * | * | * | NA | * | NA | NA | 5 |
| 19 | d'Avella, 2019 [91] | * | * | * | * | NA | * | NA | NA | 5 |
| 20 | Drimtzias, 2014 [72] | * | * | * | * | NA | * | NA | NA | 5 |
| 21 | Erşahin, 2005 [101] | * | * | * | * | NA | * | NA | NA | 5 |
| 22 | Fisher, 1998 [43] | * | * | * | * | NA | * | NA | NA | 5 |
| 23 | Fouda, 2019 [63] | * | * | * | * | NA | * | NA | NA | 5 |
| 24 | Gautier, 2012 [73] | * | * | * | * | NA | * | NA | NA | 5 |
| 25 | Gerganov, 2014 [67] | | * | * | * | NA | | NA | NA | 3 |
| 26 | Goldenberg-Cohen, 2011 [83] | * | | * | | NA | * | NA | NA | 3 |
| 27 | *Gonc, 2004* [31] | * | * | * | * | *NA* | * | *NA* | *NA* | *5* |

*(Continued)*

**Table 4.** (Continued)

| Number | Study | Selection | | | | Comparability | Outcome | | | Total number of stars |
|---|---|---|---|---|---|---|---|---|---|---|
| | | Represen-tativeness of exposed cohort | Selection of non-exposed cohort | Ascertainment of exposure | Demonstration that outcome of interest was not present at start of study | Comparability of cohorts on the basis of the design or analysis | Assessment of outcome | Was follow-up long enough for outcomes to occur | Adequacy of follow-up of cohorts | |
| 28 | Greenfield, 2015 [60] | * | * | * | * | NA | * | NA | NA | 5 |
| 29 | Haghighatkhah, 2010 [104] | | * | | * | NA | | NA | NA | 2 |
| 30 | Hakuba, 1985 [86] | | * | * | * | NA | * | NA | NA | 4 |
| 31 | Hoff, 1972 [37] | * | * | * | * | NA | * | NA | NA | 5 |
| 32 | Hoffmann, 1977 [25] | * | * | * | * | NA | * | NA | NA | 5 |
| 33 | Hoffmann, 1992 [89] | * | * | * | * | NA | * | NA | NA | 5 |
| 34 | Hoffmann, 2015 [68] | * | * | * | * | NA | | NA | NA | 4 |
| 35 | Honegger, 1999 [26] | * | * | * | * | NA | | NA | NA | 4 |
| 36 | Hoogenhout, 1984 [93] | * | * | * | * | NA | | NA | NA | 4 |
| 37 | Im, 2002 [97] | * | * | * | * | NA | | NA | NA | 4 |
| 38 | Jane jr., 2010 [64] | * | * | * | * | NA | * | NA | NA | 5 |
| 39 | Jung, 2010 [98] | * | * | * | * | NA | * | NA | NA | 5 |
| 40 | Karavitaki, 2005 [48] | * | * | * | * | NA | * | NA | NA | 5 |
| 41 | Kennedy, 1975 [39] | * | * | * | * | NA | * | NA | NA | 5 |
| 42 | Kiran, 2008 [79] | | * | * | * | NA | * | NA | NA | 4 |
| 43 | Kramer, 1960 [38] | * | * | * | * | NA | * | NA | NA | 5 |
| 44 | Lee, 2008 [108] | | * | * | * | NA | | NA | NA | 3 |
| 45 | Lena, 2005 [24] | * | * | * | * | NA | * | NA | NA | 5 |
| 46 | Leng, 2012 [58] | * | * | * | * | NA | * | NA | NA | 5 |
| 47 | Merchant, 2002 [65] | * | * | * | * | NA | * | NA | NA | 5 |
| 48 | Mohd-Ilham, 2019 [7] | * | * | * | * | NA | * | NA | NA | 5 |
| 49 | Mottolese, 2001 [33] | * | * | * | * | NA | | NA | NA | 4 |
| 50 | Nielsen, 2013 [103] | * | * | * | * | NA | * | NA | NA | 5 |
| 51 | Ohmori, 2007 [28] | * | * | * | * | NA | | NA | NA | 4 |
| 52 | Ono, 1996 [87] | | * | * | * | NA | * | NA | NA | 4 |
| 53 | Pascual, 2018 [55] | * | * | * | * | NA | | NA | NA | 4 |
| 54 | Patel, 2017 [56] | * | * | * | * | NA | | NA | NA | 4 |
| 55 | Puget, 2007 [14] | * | * | * | * | * | | NA | NA | 5 |
| 56 | Qi, 2012 [77] | * | * | * | * | NA | | NA | NA | 4 |

(Continued)

**Table 4.** (Continued)

| Number | Study | Selection | | | | Comparability | Outcome | | | Total number of stars |
|---|---|---|---|---|---|---|---|---|---|---|
| | | Represen-tativeness of exposed cohort | Selection of non-exposed cohort | Ascertainment of exposure | Demonstration that outcome of interest was not present at start of study | Comparability of cohorts on the basis of the design or analysis | Assessment of outcome | Was follow-up long enough for outcomes to occur | Adequacy of follow-up of cohorts | |
| 57 | Quon, 2019 [57] | * | * | | * | NA | * | NA | NA | 4 |
| 58 | Rath, 2012 [96] | * | * | * | * | NA | * | NA | NA | 5 |
| 59 | Richmond, 1980 [61] | | * | * | * | NA | * | NA | NA | 4 |
| 60 | Salunke, 2016 [80] | | * | | * | NA | * | NA | NA | 3 |
| 61 | Sankhla, 2015 [27] | | * | * | * | NA | | NA | NA | 3 |
| 62 | Shammari, 2012 [100] | * | | | | NA | | NR | NA | 1 |
| 63 | Shi, 2017 [78] | * | * | * | * | NA | * | NA | NA | 5 |
| 64 | Sogg, 1977 [109] | * | * | | * | NA | * | NR | NA | 4 |
| 65 | Stahnke, 1984 [29] | * | * | * | * | NA | * | NA | NA | 5 |
| 66 | Suharwardy, 1997 [9] | * | * | * | * | NA | * | NA | NA | 5 |
| 67 | Synowitz, 1977 [69] | * | * | * | * | NA | * | NA | NA | 5 |
| 68 | Tamasauskas, 2014 [105] | * | * | * | * | NA | * | NA | NA | 5 |
| 69 | Tan, 2017 [42] | * | * | * | * | * | * | NA | NA | 6 |
| 70 | Taphoorn, 2000 [94] | * | * | * | * | NA | * | NA | NA | 5 |
| 71 | Taylor, 2012 [74] | | * | * | * | NA | | NA | NA | 3 |
| 72 | Thomsett, 1980 [32] | * | * | * | * | NA | * | NA | NA | 5 |
| 73 | Tomita, 2005 [66] | * | * | * | * | NA | * | NA | NA | 5 |
| 74 | Villani, 1997 [92] | * | * | * | * | NA | * | NA | NA | 5 |
| 75 | Vries de, 2003 [85] | | * | * | * | NA | | NA | NA | 3 |
| 76 | Wan, 2018 [10] | * | * | * | * | NA | * | NA | NA | 5 |
| 77 | Weiss, 1989 [59] | * | * | * | * | NA | * | NA | NA | 5 |
| 78 | Wijnen, 2017 [95] | * | * | * | * | NA | * | NA | NA | 5 |
| 79 | Winkfield, 2011 [34] | * | * | * | * | NA | | NA | NA | 4 |
| 80 | Yamada, 2018 [88] | * | * | * | * | NA | * | NA | NA | 5 |
| 81 | Yano, 2016 [19] | * | * | * | * | NA | | NA | NA | 4 |
| 82 | Yu, 2015 [46] | | * | * | * | NA | * | NA | NA | 4 |
| 83 | Zhang, 2008 [30] | * | * | * | * | NA | | NA | NA | 4 |
| 84 | Zhou, 2009 [75] | | * | * | * | NA | * | NA | NA | 4 |

Studies in italics indicate studies retrieved by reference screening.

* The study met an item of the NOS.

NA: Not applicable, i.e. items do not apply to the research question and design of this review.

supradiaphragmatic CP (2 preinfundibular, 1 preinfundibular and suprasellar, 1 retroinfundibular) and 4 infradiaphragmatic CP (3 intra-suprasellar, 1 intra-para-suprasellar).

Nielsen (2012) reported 10 parasellar CP (0.5%) [103]. Erşahin reported 4 retroclival CP (0.2%) [101]. Lena (2005) [24] and Pascual (2019) [55] reported 31 infundibulo-tuberous CP (1.6%) in 31 children (1.6%). In a study by Taphoorn (2008) one of three CP was located in the foramen intraventriculare (0.05%) [94]. In a study by Erşahin (2005) one of 87 CP (0.05%) was located in the anterior cranial fossa and 3 CP (0.2%) were located in the temporal fossa (2 also with posterior cranial fossa extension) [101]. In a study by Patel (2017) one of 10 CP was located extracranial, infrasellar, in the nasal cavity and the sphenoid sinus (0.05%) [56].

## Tumour subtypes

Information about histological tumour subtype was available for 9 of 84 studies (Table 2) [6, 43, 55, 61, 67, 70, 78, 87, 105]. Adamantinous CP was present in 675 of 685 children (98.5%). Nine of 685 children (1.3%) had squamous CP [6] and one child (0.15%) had a papillary CP [105].

## Visual impairment

Of the 84 studies eligible for data extraction, in 56 studies authors provided the total number of patients in whom visual function was impaired (Table 3). For these studies, visual impairment was described in 1041 of 2071 children (50.3%) with CP at diagnosis. Authors used different terms to describe visual impairment, for instance 'visual impairment', 'visual defects', 'vision loss' and 'visual complaints'. If a definition for impaired visual function was provided by the authors, this is shown in Table 3. Twenty-eight studies did not mention the total number of children with visual impairment in general, nevertheless data about one or more subdomains of visual function (visual acuity, visual field, fundoscopy or orthoptic examination) was available for these studies. Sixty-two studies reported about visual disturbance as an anamnestic symptom at diagnosis in 1135 of 2267 (50.0%) children with a CP.

## Visual acuity

We identified 53 studies describing VA in children with CP at diagnosis (Table 3). Authors used different definitions and grading systems to describe VA. Four authors described the applied VA testing method, namely by Snellen test [9, 52], LogMAR charts [72] or preferential looking charts [10, 72]. Seven studies explicitly reported about best corrected VA (BCVA) instead of VA [6, 7, 60, 72, 83, 84, 95]. Authors of the other 46 studies did not describe whether they used BCVA or uncorrected VA (UCVA). The VA testing methods and definitions are shown in Table 3.

Decreased VA was reported in 546 of 1321 tested children (41.3%). Five studies reported about combined VF and VA data, therefore it was impossible to extract VA of these studies [24, 59, 60, 73, 96]. Furthermore, Taylor (2012) only reported about reduced VA without providing the number of patients [74]. Seven studies expressed decreased VA in eyes instead of in patients. Ammirati (1988) reported decreased VA in 3 of 4 eyes: VA 20/100 in one eye, VA 20/200 in one eye and perception of light (PL) in one eye [71]. Behari (2003) reported for a total of 4 eyes a VA of 6/24 in one eye, a VA of 6/18 in two eyes and a VA of 6/12 in one eye [81]. Drimtzias (2014) described deceased VA in 11 of 20 patients (40 eyes in total), with mild-moderate visual loss in 8 of 40 eyes, severe visual loss in 13 of 40 eyes and a normal VA in 19 of 40 eyes [72]. Mohd-Ilham (2019) reported the BCVA in 22 eyes, which was $\geq$ 6/6-6/12 in 13 eyes, 6/15-6/60 in 3 eyes and < 6/60 in 6 eyes [7]. In a study by Sogg (1977), two children both had decreased VA (20/100, 20/50, 20/400 and 20/200) [109]. Salunke (2016) described decreased

VA in two children, with VA 6/18 in 2 eyes, VA 6/24 in one eye and PL plus in one eye [80]. Suharwardy (1997) reported decreased VA in 10 eyes, namely VA of 6/24 in 2 eyes; VA of 6/12 in 2 eyes; VA of 6/9 in one eye; VA of 6/6 in 2 eyes; VA of 1/60 in one eye; no PL in one eye and hand motion in one eye [9].

Twenty-nine studies described decreased VA in one or both eyes without giving any further details about the degree of VA reduction in 365 of 831 children (43.9%). Visual loss was found in 31 of 68 children (45.6%) [39, 43, 57, 86, 104]. Blindness in one or two eyes with or without PL was present in 71 of 515 children with CP (13.8%) [10, 14, 24, 61, 77, 89, 96, 99, 101, 103, 104, 106]. Ali (2013) [54] and Quon (2019) [57] described blurry vision in 3 of 23 patients (13.0%). Loss of central VA was reported in 12 of 18 children by Chamlin (1955) [52].

Multiple studies described VA by using VA scales. In a study by Chen (2003) 6 patients had a VA < 6/12 (35.3%) and 10 patients had a VA ≥ 6/12 (58.8%) [6]. In two studies, 10 of 55 patients had a VA of ≤ 20/200 in one or both eyes (18.2%) [60, 84]. Kiran (2008) [79] reported VA of 6/9 in one of 2 patients and Kramer (1960) [38] VA of 6/60 in one of 6 patients. Tamasauskas (2014) described two children with a VA of 5/10 and 1/1000 [105]. Two children with CP in a study by Anderson (1989) had respectively a VA of 20/25 and counting fingers (CF) at 1 foot, and a VA of 20/300 and CF at 2 feet [53]. Finally, Synowitz (1977) presented VA data of 3 CP patients: one patient had no VA defects; one patient had only PL and the last patient could CF with his right eye at 1 m and could see hand movements with his left eye at 50 cm [69].

In summary, different grading systems and testing methods were used to report about decreased VA in 41.3% of children, with no specification of VA reduction in 43.9%. Blindness in one or both eyes was reported in 13.8% of children.

## Visual fields

A total of 46 studies provided data about visual field testing in children with CP (Table 3). Nine authors described which VF test is performed in their study, namely the Humphrey Field Analyzer [6, 7, 10, 84], Goldmann perimetry [6, 9, 10, 72, 95, 109], Bjerrum screen [6, 39], confrontation method [7, 10] and/or the red test and large white test objects [9].

Visual field defects were reported in 426 of 1111 tested children (38.3%). Mohd-Ilham (2019) reported about VF per eye instead of per patient: temporal hemianopia was found in 5 of 22 eyes, scotoma in 3 of 22 eyes, quadrantanopia in 2 of 22 eyes and a constricted VF in 1 of 22 eyes [7]. Five studies reported VF data together with VA data, therefore VF data from these studies could not be extracted [24, 59, 60, 73, 96]. In 8 studies VF defects were reported in 121 of 400 children (30.3%) without providing descriptions of the VF defects [39, 63, 64, 90, 92, 95, 101, 106]. In nine studies a VF defect (no further specification), reduction or loss was present in 82 of 320 children (25.6%) [24, 29, 32, 43, 48, 66, 82, 89, 103]. The remaining studies reported the type of the VF defect in detail. Bitemporal hemianopia was reported in 98 of 332 patients (29.5%) with pertinent data [6, 9, 25, 31, 38, 48, 52, 57, 61, 71, 72, 80, 84, 86, 89, 91, 94, 96, 99, 102, 105, 107, 109]. Twenty-three of 177 children (13.0%) were diagnosed with an unitemporal hemianopia [6, 25, 53, 61, 84, 94, 96, 102, 105]. For 11 of 33 children (33.3%) it was not specified whether their temporal hemianopia was uni- or bilateral, these are reported as having a temporal hemianopia [9, 53, 91, 107]. Zhou (2009) found hemianopia in one of 5 children (20%) [75]. Quadrantopia was described in 2 of 23 children (8.7%) [84, 91]. Richmond (1980) described the presence of a scotoma in one of 21 children [61]. Kramer found loss of right nasal field in one of 6 patients [38]. Impaired VF was reported in one of 4 patients by Boekhoff (2019) [70]. Artero (1984) found homonymous defects in 5 of 18 patients (27.8%) [107]. Suharwardy (1997) described an asymmetric binasal loss with enlarged blind spots and

a supero-temporal loss in one of 5 patients [9]. Concentric narrowing of the VF or only central VF was reported in 6 of 41 patients (14.6%) [99, 102]. Cherninkova (1990) reported 'other defects' for one of 21 patients with VF defects among their patients [102].

Despite the fact that 8 studies did not specify the VF defects in 30.3% of children with CP, uni- and/or bitemporal hemianopia is the most frequent VF defect in 132 of 542 children (24.4%).

## Fundoscopy

In 37 studies fundoscopy was performed (Table 3). Fundoscopic abnormalities were reported in 520 of 1601 examined children (32.5%). Papilledema (uni- or bilateral), also mentioned as optic disc or nerve edema, was present in 254 of 986 patients (25.8%). Optic atrophy or pallor was reported in 239 of 534 (44.8%). Weiss (1989) reported about asymptomatic papilledema or optic atrophy in 5 of 31 patients (16.1%) [59]. Optic nerve compression was found in 1 of 16 patients (6.25%) by Quon (2019) [57]. Al-Mefty (1985) reported about the presence of the Foster-Kennedy syndrome in 3 of 15 patients (20%) [99]. An abnormal ocular fundus without further specificity was reported by Artero (1984) in 19 of 24 patients (79.2%) [107].

Summarizing this, fundoscopic abnormalities were reported in 32.5% of children. Among these, papilledema (25.8%) and optic nerve atrophy or pallor (44.8%) were the most common fundoscopic abnormalities.

## Orthoptic examination

Twenty-nine studies provided data about orthoptic examination at diagnosis in children with CP (Table 3) In these studies, orthoptic abnormalities were reported in 163 of 1304 children (12.5%) with CP at diagnosis. Taylor (2012) was the only study that did not provide numbers of children in whom an orthoptic abnormality was found, they only mentioned nystagmus and strabismus as the orthoptic abnormalities seen among their study participants [74]. Fourty-three of 296 children experienced diplopia (14.5%) [7, 31, 54, 55, 60, 66, 75, 83, 84, 89, 94, 107], 21 of 127 children (16.5%) were diagnosed with strabismus (also called squint by some studies) [6, 7, 39, 66, 83, 84, 96] and in 22 of 331 children (6.6%) nystagmus (monocular, seesaw, horizontal pendular or rotatory) was seen during orthoptic examination [25, 38, 39, 69, 83, 84, 89–102].

Sixth nerve deficits or palsy were present in 15 of 220 patients (6.8%) [55, 59, 81, 84, 90, 101] and other cranial nerve deficits or palsies in 22 of 159 patients (13.8%) [32, 46, 90, 102]. Proptosis was reported in one of 16 children (6.3%) with CP by Quon (2019) [57]. Four studies mentioned a relative afferent pupillary defect (RAPD) in 30 of 46 children (65.2%) [6, 7, 9, 84]. Wan (2018) reported for diplopia and strabismus together in 7 of 59 patients (11.9%) [10]. In a study by Erşahin (2005) diplopia and squint were reported together which was seen in 7 of 87 (8.0%) patients [101].

The overall findings in children with orthoptic abnormalities (12.5%) showed diplopia in 14.5%, strabismus in 16.5% and nystagmus in 6.6% of the children.

## Other vision related abnormalities

Apart from the abovementioned ophthalmological findings, some studies have described other vision related abnormalities as well (Table 3). Colour vision defects were reported by Mohd-Ilham (2019) in 4 of 11 patients (18.2%) [7]. Nielsen (2012) described ophthalmoplegia in 5 of 31 patients (16.1%) [103]. Right amaurosis was reported by d'Avella (2019) [91] in 1 of 8 patients (12.5%) and Chamlin (1955) [52] reported ptosis in 1 of 18 patients (0.05%). These

vision related abnormalities were not the main focus of our study and were not analysed and/or reported in any of the other studies included in our systematic review.

## Discussion

Our review was designed to provide a detailed overview of the currently available evidence about visual function in children with CP at diagnosis. To the best of our knowledge, this is the first review that systematically describes the visual function in subtoptics like VA, VF, fundoscopy and orthoptic examination. We included 84 studies, with 56 studies explicitly providing data about visual impairment in general, and 55 studies providing specific data about VA and/or VF. We found a high rate of visual impairment in children with CP at time of diagnosis (50.3%). Considerable rates were also reported for decreased VA (41.3%) and VF loss (38.3%). Papilledema (25.8%) and optic nerve atrophy (44.8%) were common fundoscopic findings in our review. The most common abnormalities in orthoptic examination (12.5%) were strabismus, diplopia and cranial nerve deficits. These findings are in agreement with several nonsystematic reviews of Bogusz (2018), who concluded that more than 50% of children with CP had visual impairment at diagnosis [110], and with Müller (2008) who described visual impairment, VF defects, papilledema and optic atrophy in respectively 62–84%, 36%, 20–35% and 35–45% of children with CP [4]. Drapeau (2019) described even higher rates for decreased VA and VF defects, namely in 70–80% of children with CP. In particular, Drapeau (2019) reported bitemporal hemianopia and papilledema in respectively 50% and 20% of children with CP [111].

The presented data in our review supports the importance of awareness in doctors for the fact that CP commonly induce visual impairment in children, as well as the importance of ophthalmological examination at diagnosis. Visual impairment due to damage of the optic nerves, optic chiasm and visual pathways often results in lifelong effects for children and their family, by affecting domains including childhood development, education, employment and self-perception [12, 112–114]. Visual problems may be reversible in early stages of visual impairment. Therefore, timely monitoring of visual function and early detection of visual impairment in children with CP is of major importance to preserve visual function and provide adequate treatment [12, 13, 17, 18, 20]. In children with irreversible visual impairment, timely referral for visual rehabilitation may reduce the adverse effects of visual impairment on health and/or vision related quality of life [115, 116]. The impact of an impaired visual function on quality of life has also been reported in children with visual impairment with ophthalmological origin, for example, children who suffer from glaucoma and cataract [117, 118].

Moreover, visual impairment has been reported as one of the factors that may lower the level of physical activity [119]. Especially in children with cranioharyngioma in whom hypothalamic damage can be severe, resulting in endocrine deficiencies and obesity, physical activity is crucial (40–50%) [119–123]. Both visual impairment and severe obesity negatively affect quality of life in childhood CP survivors [119, 124, 125].

### Limitations of the included studies

Overall there was moderate-quality evidence for the presence of visual impairment in children with CP at diagnosis in our review. Although there were serious limitations to the data due to e.g. retrospective design of the studies, moderate risk of bias for some included studies and potential publication/reporting bias, the overall quality of evidence was raised by the number of included studies, study sizes, availability of confirmatory evidence and representativeness of study patients. Nevertheless, there are some issues that need to be discussed. First, different terminology was used to describe tumour locations and no concrete insights were given in the relationship between tumour location and visual loss. Therefore, we were not able to relate a

more suprasellar tumour location involving the optic chiasm with the type and degree of visual loss. Second, no standardized ophthalmological examination was performed in a large proportion of included studies, and if performed, there was no uniformity in testing methods between the studies. Visual acuity and VF were described with different definitions and cut-off values per study, which makes grouping of results difficult. In the absence of standardized ophthalmological examination, it could be questioned whether we can presume that those children reported without visual impairment really have a normal visual function. In addition, performing reliable VA and VF testing in young and non-cooperative children is often very complicated [126, 127]. Therefore, it is likely that these data were frequently missing in the included studies. Both of these issues, no standardized ophthalmological examination and difficulties with reliable VA and VF testing, could be reasons for underreporting of visual impairment in our review. Additionally, we were not able to compare the feasibility of different VA and VF testing methods for different age groups, because only a few studies provided information about the used tests and in these studies the authors often did not specify which testing methods were used for the different age groups. Furthermore, study authors reported the number of children per abnormality found by either fundoscopy or orthoptic testing. For this reason, the exact number of children with fundoscopic or orthoptic abnormalities is unclear, because one child might have more than one fundoscopic or orthoptic abnormality. Finally, authors used different cut-off values for the age limit of children. We initially planned to include studies only when patients were aged between 0 and 18 years. However, during study selection we encountered multiple studies still referring to patients as 'children' when aged < 24 years. We decided to include those studies as well, aiming to provide an extensive overview of visual function in children with CP at diagnosis. Nevertheless, heterogenity in age ranges for children across studies may lower their comparibility.

## Strengths and limitations of this systematic review

The findings of the present systematic review should be interpreted by its strengths and limitations. We planned this review a priori and registered our review in PROSPERO with clearly-defined selection criteria. We conducted a comprehensive literature search and reviewed all reference lists of included studies. Two reviewers independently of each other performed the literature screening, data extraction and risk of bias assessment. In this way we retrieved and summarized the visual data of 3531 children with newly diagnosed CP. We also encountered some possible limitations for the methodology of our review. By screening references of included studies, we identified a relatively high number of additional studies eligible for inclusion (n = 15). Therefore, it might be possible that in our search we missed studies, though our cross reference search would in that case have identified those articles. Furthermore, full-text articles of 117 potential relevant abstracts found by our search in the electronic databases were not available despite searching the Utrecht University Library, Sci-Hub and contacting the corresponding author by mail and/or ResearchGate. Possible reasons for this could be that the abstracts are dated (we did not use a publication date filter) or that the full text did not exist. Lastly, we did not review visual function at follow-up as we initially planned for in our PROSPERO registration. This was because many authors within our study selection, described visual follow-up data only in patients who received tumour treatment.

## Recommendations

For future research, it is relevant to investigate the visual function at diagnosis and during long-term follow-up of childhood CP in response to surgery, radiotherapy and other treatment strategies, first by systematically reviewing the literature as well as in prospective collaborative studies. This will provide insight in risks and benefits of treatment regarding vision in children

with CP, for professionals, patients and their caregivers. Furthermore, future studies should focus on reliable ophthalmological testing methods for young and non-cooperative children. As we have shown, the majority of studies did not report the methods they used for ophthalmological testing. Therefore, unfortunately, we were not able to compare feasibility of testing methods for different age groups due to lacking data. For future studies, it is important that all studies must use the correct testing methods for VA and VF and report these as such in the paper. Additionally, optical coherence tomography with analysis of the retinal layers might be applied as objective testing method in addition to VA and VF testing [84, 128–130].

## Conclusion

Children diagnosed with CP have at least 50% risk of visual impairment at diagnosis regarding VA, VF, fundoscopy and/or orthoptic examination. Complete structured evaluation of visual function at diagnosis should be performed routinely in all children diagnosed with craniopharyngioma. However, large, well designed studies with standardized ophthalmological examination and uniform reporting with grading are needed to gain more insight in the visual function of these patients at diagnosis, after therapeutic interventions and during follow-up.

## Supporting information

**S1 Appendix. Search strategies for electronic databases.**
(DOCX)

**S2 Appendix. PRISMA checklist.**
(DOC)

**S3 Appendix. Systematic research protocol.**
(PDF)

## Author Contributions

**Conceptualization:** Myrthe A. Nuijts, Nienke Veldhuis, Inge Stegeman, Hanneke M. van Santen, Giorgio L. Porro, Saskia M. Imhof, Antoinette Y. N. Schouten–van Meeteren.

**Data curation:** Myrthe A. Nuijts, Nienke Veldhuis.

**Formal analysis:** Myrthe A. Nuijts, Nienke Veldhuis.

**Methodology:** Myrthe A. Nuijts, Nienke Veldhuis, Inge Stegeman, Saskia M. Imhof, Antoinette Y. N. Schouten–van Meeteren.

**Project administration:** Myrthe A. Nuijts, Nienke Veldhuis.

**Supervision:** Inge Stegeman, Hanneke M. van Santen, Giorgio L. Porro, Saskia M. Imhof, Antoinette Y. N. Schouten–van Meeteren.

**Writing – original draft:** Myrthe A. Nuijts, Nienke Veldhuis.

**Writing – review & editing:** Inge Stegeman, Hanneke M. van Santen, Giorgio L. Porro, Saskia M. Imhof, Antoinette Y. N. Schouten–van Meeteren.

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
