## [Decision Letter · Decision Letter 0]

24 Aug 2020

PONE-D-20-13068

Visual function in children with craniopharyngioma at diagnosis: a systematic review.

PLOS ONE

Dear Dr. Nuijts,

Thank you for submitting your manuscript to PLOS ONE. After careful consideration, we feel that it has merit but does not fully meet PLOS ONE’s publication criteria as it currently stands. Therefore, we invite you to submit a revised version of the manuscript that addresses the points raised during the review process.

ACADEMIC EDITOR: No Comments

We look forward to receiving your revised manuscript.

Kind regards,

Ahmed Awadein, MD, Ph.D, FRCS

Academic Editor

PLOS ONE

Journal Requirements:

Reviewers' comments:

Reviewer's Responses to Questions

**Comments to the Author**

1. Is the manuscript technically sound, and do the data support the conclusions?

Reviewer #1: Yes

Reviewer #2: Yes

2. Has the statistical analysis been performed appropriately and rigorously? 

Reviewer #1: I Don't Know

Reviewer #2: Yes

3. Have the authors made all data underlying the findings in their manuscript fully available?

Reviewer #1: Yes

Reviewer #2: Yes

4. Is the manuscript presented in an intelligible fashion and written in standard English?

Reviewer #1: Yes

Reviewer #2: Yes

5. Review Comments to the Author

Reviewer #1: The authors submitted a manuscript entitled "Visual function in children with craniopharyngioma at diagnosis: a systematic review". They systematically reviewed the literature to provide an extensive overview of the visual

function in children with craniopharyngioma at diagnosis in order to estimate the

diversity, magnitude and relevance of the problem of visual impairment. Of the 543 potentially relevant articles, 84 studies met their inclusion criteria. Visual impairment at diagnosis was reported in 1041 of 2071 children (50.3%), decreased visual acuity was reported in 546 of 1321 children (41.3%) and visual field defects were reported in 426 of 1111 children (38.3%). Other ophthalmological findings described were fundoscopic (32.5%) and orthoptic abnormalities (12.5%).

My comments to the authors are:

1- I suggest that the full title is modified to "Visual functions at diagnosis in children with craniopharyngioma: a systematic review".

2- The running title needs also to be modified to "Visual functions in childhood craniopharyngioma".

3-In the study characteristics, page 9 lines 198-199 "With regard to the included children with CP, 1236 were female and mean age ranged from 0 (39,93) to 23 years", the sentence needs re-phrasing to "As regards to the included children with CP, 1236 were females and the mean age ranged from 0 to 23 years".

4-In the study characteristics page 9, lines 209-210 "To be precise, only 10 studies primarily reported about visual function and/or long-term visual outcomes in children with CP", this sentence also needs re-phrasing to "Precisely only 10 studies reported primarily visual function and/or long term visual outcomes in children with CP".

5-In pages 28-29, lines 289-295: the term visual impairment is vague, could the authors clarify what do they exactly mean by visual impairment? if they mean visual acuity and visual field then this item should be not be mentioned separately.

6-In page 29, visual acuity paragraph the authors mentioned that 53 studies out of 84 studies described visual acuity at diagnosis and only 4 studies referred to visual acuity as BCVA. Is there an explanation for that? How do the authors justify

using UCVA and BCVA as a description for visual acuity affection in these children?.

Reviewer #2: Nuijts et al. provide a comprehensive overview of the visual function in children with craniopharyngioma at diagnosis. Their manuscript is very well written, innovative and provides important findings for the clinical care of these patients with their very rare disease.

I have only one point, which I would like to suggest for revision / additional editing: From a clinical point of view, it is important to know the rate and risk for pathological ophthalmological findings at diagnosis of craniopharymgioma in pediatric patients.

BUT it is sometimes difficult to chose the age-appropriate examination / opthalmological test to reach reliable diagnostic results especially in very young children. Could the authors check the reviewed studies, which testing methods were used (and feasible) in different age groups. Perhaps these analyses could lead to a cautious recommendation of what should be done in terms of opthalamological diagnostics at different age periods.

6. PLOS authors have the option to publish the peer review history of their article (what does this mean?). If published, this will include your full peer review and any attached files.

Reviewer #1: No

Reviewer #2: **Yes: **Hermann L. Müller

---

## [Author Response · Author response to Decision Letter 0]

7 Sep 2020

Reviewer 1: 

1- I suggest that the full title is modified to "Visual functions at diagnosis in children with craniopharyngioma: a systematic review".

- We thank the reviewer for this comment. We agree with the suggested full title. (Page 1, line 4).

2- The running title needs also to be modified to "Visual functions in childhood craniopharyngioma". 

- We agree with the suggested running title. (Page 1, line 7).

3-In the study characteristics, page 9 lines 198-199 "With regard to the included children with CP, 1236 were female and mean age ranged from 0 (39,93) to 23 years", the sentence needs re-phrasing to "As regards to the included children with CP, 1236 were females and the mean age ranged from 0 to 23 years".

- In the revised manuscript, we rephrased this sentence to ‘’With regard to the included children with CP, 1236 were females and the mean age ranged from 0 to 23 years’’. (Page 8, lines 183-185).

4-In the study characteristics page 9, lines 209-210 "To be precise, only 10 studies primarily reported about visual function and/or long-term visual outcomes in children with CP", this sentence also needs re-phrasing to "Precisely only 10 studies reported primarily visual function and/or long term visual outcomes in children with CP".

- In the revised manuscript, we rephrased this sentence. (Page 9, lines 195-196).

5-In pages 28-29, lines 289-295: the term visual impairment is vague, could the authors clarify what do they exactly mean by visual impairment? if they mean visual acuity and visual field then this item should be not be mentioned separately.

- We thank the reviewer for this comment and we agree that the term visual impairment is vague. Authors frequently reported about the total number of patients with visual impairment in general without describing results of visual acuity and visual field testing. Therefore, we decided to report this data alongside the results of visual acuity and visual field testing. We rephrased this sentence in the results section aiming to clarify what we mean by using the term ‘visual impairment’. (Page 28, lines 276-285). 

6-In page 29, visual acuity paragraph the authors mentioned that 53 studies out of 84 studies described visual acuity at diagnosis and only 4 studies referred to visual acuity as BCVA. Is there an explanation for that? How do the authors justify using UCVA and BCVA as a description for visual acuity affection in these children?.

- Unfortunately, only in seven of the included studies authors explicitly described that they used BCVA as a description for visual acuity affection. In the other studies, authors only reported ‘visual acuity’ without specifying whether they used BCVA or UCVA. In the revised manuscript, we added a sentence to clarify this and added UCVA to the list of abbreviations. (Page 29, lines 292-294, page 38, lines 511-512).

Reviewer 2: 

1. I have only one point, which I would like to suggest for revision / additional editing: From a clinical point of view, it is important to know the rate and risk for pathological ophthalmological findings at diagnosis of craniopharymgioma in pediatric patients. BUT it is sometimes difficult to chose the age-appropriate examination / opthalmological test to reach reliable diagnostic results especially in very young children. Could the authors check the reviewed studies, which testing methods were used (and feasible) in different age groups. Perhaps these analyses could lead to a cautious recommendation of what should be done in terms of opthalamological diagnostics at different age periods.

- We thank the reviewer for this comment and agree that this is very important to reliably compare results of different studies. Table 3 of our manuscript shows the ophthalmologic testing methods that were used in the different studies (if provided by the authors). Regrettably, it shows that the majority of studies did not report the methods they used for ophthalmologic testing. Therefore, unfortunately we were not able to compare feasibility of testing methods for different age groups due to lacking data. This is outlined in the discussion section (see ‘limitations of the included studies’ and ‘recommendations’). (Page 36, lines 457 – 460, page 37, lines 493-497).

---

## [Editor Report · Decision Letter 1]

18 Sep 2020

Visual functions in children with craniopharyngioma at diagnosis: a systematic review.

PONE-D-20-13068R1

Dear Dr. Nuijts,

We’re pleased to inform you that your manuscript has been judged scientifically suitable for publication and will be formally accepted for publication once it meets all outstanding technical requirements.

Kind regards,

Ahmed Awadein, MD, Ph.D, FRCS

Academic Editor

PLOS ONE
---

## [Editor Report · Acceptance letter]

22 Sep 2020

PONE-D-20-13068R1 

Visual functions in children with craniopharyngioma at diagnosis: a systematic review. 

Dear Dr. Nuijts:

I'm pleased to inform you that your manuscript has been deemed suitable for publication in PLOS ONE. Congratulations! Your manuscript is now with our production department. 

Kind regards, 

on behalf of

Dr. Ahmed Awadein 

Academic Editor

PLOS ONE